# Circuit Complexity through phase transitions:
# consequences in quantum state preparation

Sebastián Roca-Jerat,[1] Teresa Sancho-Lorente,[1] Juan Román-Roche,[1] and David Zueco[1]

[1]*Instituto de Nanociencia y Materiales de Aragón (INMA) and Departamento de Física de la Materia Condensada,*
*CSIC-Universidad de Zaragoza, Zaragoza 50009, Spain*

(Dated: October 11, 2023)

In this paper, we analyze the circuit complexity for preparing ground states of quantum many-body systems. In particular, how this complexity grows as the ground state approaches a quantum phase transition. We discuss different definitions of complexity, namely the one following the Fubini-Study metric or the Nielsen complexity. We also explore different models: Ising, ZZXZ or Dicke. In addition, different forms of state preparation are investigated: analytic or exact diagonalization techniques, adiabatic algorithms (with and without shortcuts), and Quantum Variational Eigensolvers.

We find that the divergence (or lack thereof) of the complexity near a phase transition depends on the non-local character of the operations used to reach the ground state. For Fubini-Study based complexity, we extract the universal properties and their critical exponents.

In practical algorithms, we find that the complexity depends crucially on whether or not the system passes close to a quantum critical point when preparing the state. For both VQE and Adiabatic algorithms, we provide explicit expressions and bound the growth of complexity with respect to the system size and the execution time, respectively.

**CONTENTS**

# I. INTRODUCTION

How much does it cost to generate a *target* quantum state from another *reference* state? This is a rather general question that has been discussed in quantum information for obvious reasons. In quantum computation it is desirable to obtain the result with the minimum set of gates. This number is, roughly speaking, the computational cost and it is called *circuit complexity* ($\mathcal{C}$) [1–3]. It is, let us say, the quantum analog of the concept of computational complexity in computer science. Importantly enough, this cost builds upon a concrete physical architecture, i.e the available set of gates. Therefore, $\mathcal{C}$ not only depends on the reference and target states but on the restrictions for reaching the latter. This is quite natural if one thinks of an actual quantum computer where the possible operations have restrictions. Note that, if any unitary were allowed, a simple *rotation* would achieve the goal and every quantum state would be easily prepared, so that (essentially) the complexity would be a trivial quantity. Therefore, also in analytic calculations, the path between the reference and target is restricted to a set, e.g. gaussian states [4–8] .
Beyond quantum computation, circuit complexity is a relevant concept in quantum gravity. In particular, for its consequences in holography [9–11]. For those who are not experts (like us), we can say that holography describes quantum gravity within a region of space by looking at the boundary of that region, that is described by a non gravitational theory. Then, any bulk quantity in the gravitational theory is equivalent or *dual* to another quantity in the boundary of the non gravitational theory. One of the main problems of this duality is that the volume behind the black hole horizon keeps growing for a very long time while the entanglement at the boundary saturates at much shorter times. One possible solution is to conjecture that the dual of volume is not entanglement but complexity, via the identification *Complexity = Volume*. This is because we expect that volume is an extensive quantity, while entanglement (typically) fulfils an area law. Therefore, the calculations of complexity are beyond the quantum information community and different calculations in field theories have been discussed in the literature [12, 13].

The notion of complexity ($\mathcal{C}$) is much related to the geometry of states (or operators). It is a measure of the distance between two of them. Therefore, one possible choice for $\mathcal{C}$ is finding the geodesics in the Fubini-Study metric in the projected Hilbert space for the case of pure states. For mixed states different measures have been introduced via state purification [14] or distance measures for mixed states as the Bures distance [15]. This geometric background is a powerful way to understand complexity, since it allows us to know how much it will cost to prepare a state by solving a geodesic equation. It is true, however, that the metric, in principle, can only be obtained in some cases: surely in exactly solvable models. And there we know how to prepare states. Thus, it is interesting to be able to predict the typical behaviour in general models. Here, we move in this direction.

In this article we are interested in a quite generic situation, *i.e.* when a critical point is crossed to reach the target state. In particular, we investigate what general statements about the behaviour of the circuit complexity we can make. We are not the first to calculate $\mathcal{C}$ in a quantum phase transition (QPT) [16]. Recent papers discuss exactly solvable models as the topological Kitaev, Bose-Hubbard Lipkin-Meshkov-Glick ones and conformal field theories [17–25]. Importantly enough, complexity has been shown to be a useful probe of topological phase transitions. Complementary to these calculations, in this work, we use that close to a transition point, the concept of universality emerges naturally, so we expect these universal properties to be inherited by complexity. If so, we can argue for its scaling laws or how complexity behaves regardless of model details or *even* on the particular chosen definition of complexity. In addition, we apply our theory for state preparation in quantum computers. This is a key and hard task [26]. It is within the QMA complexity class [27], roughly speaking the NP-complete analogue for quantum computers. Nevertheless, quantum computers are expected to be better than classical methods such as density functional theory [28], density normalization group [29], tensor networks [30], quantum Montecarlo [31] or even ML-inspired techniques [32], in some instances. For a recent discussion of these issues, see [33]. Heuristic quantum algorithms as adiabatic [34] or varational ones [35, 36] can outperform classical calculations and serve for the generation of quantum states as quantum data, *e.g.* phase classification [37]. Motivated by all of this, we discuss how useful the concept of complexity is and how much one can anticipate the difficulty of state preparation in variational quantum eigensolvers (VQEs) or adiabatic quantum algorithms (with and without shortcuts to adiabaticity). To challenge our theory we tackle both integrable and non-integrable models using numerical simulations and computing $\mathcal{C}$.

## A. Complexity overview

We find it convenient to discuss first the different notions of circuit complexity that we will use in this paper and the relationship between them.

### 1. Complexity à la Nielsen

The original notion of complexity is due to the works of Nielsen and collaborators [1–3]. See [13] for a recent review. Restricting ourselves to unitary operations, target and reference states are related as

$$|\psi_T\rangle = U(t,0)|\psi_R\rangle = \mathcal{T}e^{-i\int_0^t \mathcal{H}(\tau)\,d\tau}|\psi_R\rangle \ . \tag{1}$$

$\mathcal{T}$ stands for time ordering. Notice that,

$$\mathcal{H}(\tau) = i(\partial_\tau U)U^\dagger \ . \tag{2}$$

A Cost function is formally defined as:

$$\mathcal{C}_{\mathrm{N}} := \min_{\{U\}} \int_0^t d\tau \ F[U, \dot{U}] \tag{3}$$

with $F$ some functional fulfilling some basic properties as continuity, homogeneity ($F[U, \lambda \dot{U}] = \lambda F[U, \dot{U}]$ for $\lambda \geq 0$), positivity and the triangular inequality [1]. If, in addition to these, smoothness is assumed and the Hessian of $F$ as a function of $U$ is strictly positive, the functional is a Finsler metric. Thus, $\mathcal{C}_{\mathrm{N}}$ is nothing but the geodesics. The suffix N stands for Nielsen.

Being a little more explicit, we can write that the evolution is given by

$$\mathcal{H}(\tau) = \sum_n Y^{(n)}(\tau)O_n \ . \tag{4}$$

with $O_n$ some operators and $Y^{(n)}(\tau)$ parameters. A usual functional is then given by,

$$F_k(\tau) \sim \left( \sum_n |Y^{(n)}(\tau)|^k \right)^{1/k} \ . \tag{5}$$

If we restrict ourselves to two level systems (qubits), $F_k(\tau)$ is a natural distance in $SU(2^n)$, such that $d = \int_0^t d\tau \ F_k(\tau)$, Cf. with Eq. (3). What has been explained so far is the continuous version of complexity, that provides a lower bound for the number of gates needed to approximate $|\psi_T\rangle$ from $|\psi_R\rangle$ [1]. The *discrete* version of $\mathcal{C}_{\mathrm{N}}$ can be computed introducing the functional (using the same notation as in the original [1]):

$$F(\tau) = \sqrt{\sum_\sigma{}' h_\sigma(\tau)^2 + p^2 \sum_\sigma{}'' h_\sigma(\tau)^2} \tag{6}$$

where the Hamiltonian in this case is $\mathcal{H}(\tau) = \sum_\sigma' h_\sigma(\tau)\sigma + \sum_\sigma'' h_\sigma(\tau)\sigma$. In the first sum, $\sigma$ ranges over all possible one- and two-body interactions, that is, over all products of either one or two qubit gates. In the second sum, instead, the sum is over other tensor products of Pauli matrices and the identity. The factor $p > 0$ penalizes three, four, ... -body interactions. All put together, finding the geodesics in the continuum version is a good estimate of the resources needed to prepare a state.

At this point, we think it is necessary to emphasise something. If any unitary is possible, the complexity has a narrow utility, since its value is given by $\mathcal{C} = \arccos(|\langle\psi_R|\psi_t\rangle|)$, *i.e.* of the order of one (it doesn't matter which state reference and destination are chosen). This can be verified by noting that the target state can always be written as $|\psi_T\rangle = \cos\theta|\psi_R\rangle + e^{i\gamma}\sin\theta|\psi_R^\perp\rangle$ with $\langle\psi_R|\psi_R^\perp\rangle = 0$. A rotation in the subspace generated by $\{|\psi_R\rangle, |\psi_R^\perp\rangle\}$ does the job. Therefore, some restrictions on the possible unitaries or Hamiltonian (4) will be imposed. We will discuss this point in some depth later.

--------

[1] Notice that due to homogeneity, w.l.o.g. we can always set $t = 1$.

## 2.  Circuit Complexity from the Fubini-Study metric

The functionals $F$ discussed so far, see Eqs. (4) and (5), are not unique. Others can be chosen satisfying continuity, homogeneity, positivity and the triangular property. We want to discuss next another possibility where the distance between the reference and target states is given by the Fubini-Study metric. Originally proposed for Quantum Field Theories in [5], we prefer to study it here from a quantum information perspective. Let us time-slice the unitary (1) such that

$$|\psi_T(\lambda)\rangle = U_\lambda(t, t_{N-1})...U_\lambda(t_1, t_0)|\psi_R(\lambda)\rangle \;\rightarrow\; |\psi(\lambda; t_n)\rangle = U(t_n, t_{n-1})|\psi(\lambda; t_{n-1})\rangle \tag{7}$$

We have assumed that the unitaries and so the wave functions depend on the parameters $\lambda$. Then, for sufficiently small time step $\delta\tau := t_n - t_{n-1}$, the fidelity between two *contiguous* states is

$$\mathcal{F}_{n,n+1} \equiv |\langle\psi(\lambda; t_n)|\psi(\lambda; t_{n-1})\rangle| = 1 - \chi_F\,\delta\tau^2 + \mathcal{O}(\delta^4) \tag{8}$$

where $\chi_F$ is denoted the fidelity susceptibility [38–44], see Ref. [45] for a review. Interestingly enough, we can relate $\chi_F$ with the geometric tensor, in fact [Cf. Eq. (11)]

$$\chi_F = g_{\mu\nu}\dot{\lambda}^\mu\dot{\lambda}^\nu \tag{9}$$

with [5, 46]

$$g_{\mu\nu} = \mathrm{Re}\,(T_{\mu\nu})\;. \tag{10}$$

Here, $T_{\mu\nu}$ is the quantum geometric tensor which is nothing but the Fubini-Study metric (FSM) on the $CP^n$ manifold, namely:

$$T_{\mu\nu} = \langle\partial_{\lambda_\mu}\psi|P|\partial_{\lambda_\nu}\psi\rangle \tag{11}$$

with $P = 1 - |\psi\rangle\langle\psi|$ [2].
Another useful way of writing the metric tensor is as follows. Using a formal Taylor expansion for the states, the metric tensor can be written as,

$$g_{\mu\nu} = \frac{1}{2}\left(\langle\partial_\mu\psi|\partial_\nu\psi\rangle - \langle\partial_\mu\psi|\psi\rangle\langle\psi|\partial_\nu\psi\rangle + \text{c.c.}\right) \tag{12}$$

Setting now in the Hamiltonian (4), $\dot{\lambda}_\nu \equiv Y^{(\nu)}$ then $|\partial_\nu\rangle = O_\nu|\psi\rangle$, it is straightforward to see that [6],

$$g_{\mu\nu} = \frac{1}{2}\langle\psi\,|\{O_\mu, O_\nu\}|\,\psi\rangle - \langle\psi)\,|O_\mu|\,\psi\rangle\,\langle\psi\,|O_\nu|\,\psi\rangle = \frac{1}{2}|\psi|\langle\{O_\mu - \langle O_\mu\rangle_\lambda, O_\nu - \langle O_\nu\rangle_\lambda\}|\,\psi\rangle\,, \tag{13}$$

*i.e.* the fluctuations of the Hamiltonian operators $O_\nu$.

Using the fact that $1 - \mathcal{F}_{n,n+1}$ is a distance, thus satisfying the properties we imposed for the $F$-functional, we have that we can understand $\mathcal{C}$ as the distance defined through the Fubini-Study metric:

$$\mathcal{C}_{\mathrm{FS}} := \min_{\{U\}}\int_0^t \sqrt{g_{\mu\nu}\dot{\lambda}^\mu\dot{\lambda}^\nu}\,d\tau\;. \tag{14}$$

The suffix stands for Fubini-Study metric and the notion of distance is quite explicit. This is an alternative definition to that given by Eq. (3) that has some remarkable properties. The first one is that knowing the metric tensor the geodesics can be found, at least in principle, by solving the differential equation:

$$\frac{d^2\lambda^\mu}{d\tau^2} + \Gamma^\mu_{\nu\rho}\frac{d\lambda^\nu}{d\tau}\frac{d\lambda^\rho}{d\tau} = 0 \tag{15}$$

Here, $\Gamma$ are the Christoffel symbols:

$$\Gamma^\mu_{\nu\rho} = \frac{1}{2}g^{\mu\xi}\left(\partial_\rho g_{\xi\nu} + \partial_\nu g_{\xi\rho} - \partial_\xi g_{\nu\rho}\right)\;. \tag{16}$$

The second property of $\mathcal{C}_{\mathrm{FS}}$ is that, from its relation to the fidelity between states, $\mathcal{F}$, its properties close to a QPT can be used when discussing the complexity, $\mathcal{C}$, see also Eq. (13).

---

[2] Notice that the imaginary part of $T$ is nothing but the Berry phase.

### 3.  Some remarks comparing both approaches

The Nielsen complexity estimates the minimum number of gates required to transform an initial reference state to a target state. It operates by defining a set of available gates and a metric in the space of quantum circuits or unitary transformations. The optimization process within this space aims to minimize the number of gates. The Nielsen approach is operational, providing an estimation of the minimum number of operations based on the reference and target states, along with the available gates. On the other hand, the Fubini-Study approach considers a manifold of quantum states and defines the cost as the distance between pure quantum states using fidelity, as given in equations (8) and (17). In this sense, the Fubini-Study metric serves as a geometric measure, while the Nielsen complexity directly relates to the practical cost in a quantum computer.

In practical terms, there are notable differences between the two approaches. The Fubini-Study metric assigns a variable cost to specific gates based on the states they act on, while Nielsen assigns a fixed cost to each gate. Additionally, the Nielsen complexity may involve degenerate operations that leave the state unchanged (e.g., adding a global phase), resulting in a higher dimensionality of the space of unitaries. Different results are obtained using these approaches, depending on the specific application. Each form has its preferred use case. The geodesic equations provided by the Fubini-Study metric make it suitable for analytical calculations, while the Nielsen complexity is more relevant for cost estimation in quantum computation and state preparation. However, in some cases, both methods yield identical results, such as in the preparation of Gaussian fundamental states [6].

### B.    Main results and manuscript organization

For the exactly solvable systems that we discuss in this work, we find that $\mathcal{C}_{FS} \geq \mathcal{C}_N$ when crossing a phase transition. We understand this inequality as a consequence of the fact that the unitary space is larger, see previous subsection I A 3. In any case, $\mathcal{C}$ does not diverge at the critical point, its derivative does. For $\mathcal{C}_{FS}$ we can characterize this divergence and its critical exponents in rather general circumstances. Let us remark, again, that throughout the paper we focus on the extensive part of $\mathcal{C}$. Two models are studied in detail, namely the one-dimensional quantum Ising and Dicke models.
After this general discussion, we focus on calculating the complexity when preparing a fundamental state in a quantum computer. Here, obviously, we compute $\mathcal{C}_N$ in its discrete version. We explore two algorithms in detail. First, we discuss the circuit complexity in adiabatic algorithms with and without shortcuts to adiabaticity. We focused our study on one-dimensional spin lattices of different sizes. In this investigation, we found that using shortcuts does not significantly alter the complexity $\mathcal{C}_N$. However, we demonstrated that $\mathcal{C}_N \sim \sqrt{L} \times T$, where $L$ represents the system size, and $T$ is the total time required to achieve a fixed fidelity, $\mathcal{F}$, with the exact ground state (in our case, $\mathcal{F} = 0.9$ was chosen). Thus, the complexity inherits the behavior of $T$ close to a Quantum Phase Transition (QPT). Specifically, $T$ is bounded by $\Delta^{-2}$, where $\Delta$ represents the minimum gap between the ground state and the first excited state in the adiabatic algorithm.

Then, we discuss the circuit complexity using VQEs. These algorithms are variational and do not need to cross the critical point even if the reference and target are in different phases. In such a case, $\mathcal{C}_N$ is not necessarily aware of the QPT. On the other hand, if the target state is close enough to a phase transition, also in VQEs, the complexity grows. Importantly, we provide an explicit formula for $\mathcal{C}_N$, and by combining it with the correlation length generated using local Variational Quantum Eigensolver (VQE) ansatzs, we can show that $\mathcal{C}_N \gtrsim L^{3/2}$. Therefore, this scaling poses challenges for our numerical capabilities, explaining the difficulties in finding reliable solutions around Quantum Phase Transitions (QPTs) when simulating the action of a VQE.

The rest of the manuscript is organized as follows. In the next section, II, we discuss the relation between circuit complexity, in this case $\mathcal{C}_{FS}$ from Eq. (14), and the geometry of quantum states that allows extracting the critical exponents for the derivative of $\mathcal{C}$. This is our first result that emphasizes that through phase transitions $\mathcal{C}_{FS}$ has universal properties. In section III we perform explicit calculations for $\mathcal{C}_{FS}$ and $\mathcal{C}_N$ in two solvable systems, namely the one dimensional XY-anisotropic and Dicke models. We extract the critical exponents. Then, in section IV, we perform numerical simulations where $\mathcal{C}_N$ is computed in two types of algorithms: variational and adiabatic ones. Concretely we benchmark with exactly solvable models as the Ising model, and we complement our discussion with non-integrable Hamiltonians as the ZZXZ model. Lastly, we discuss these results and conclude the paper in V. Some technical issues are left for the Appendices. The code used to obtain the numerical results is available upon request.

## II. COMPLEXITY AND THE GEOMETRY OF STATES CLOSE TO A QUANTUM PHASE TRANSITION.

In this section, we discuss general aspects for the complexity close to a QPT. To be as general as possible, we find it convenient to focus on $\mathcal{C}_{\mathrm{FS}}$, Eq. (14). Within this geometric formalism, we see that, in general, the complexity is finite, but not its derivative, which can diverge when crossing a QPT. We study its finite size scaling obtaining the corresponding critical exponents.

### A. Complexity and its derivative when crossing a QPT

We have already argued in section I A 1 that if we are allowed to use any unitary, $\mathcal{C}$ is of the order of one. In the literature, several unitary restrictions have been used: considering one and two qubit gates or considering gaussian states when moving from reference to target states. In this subsection, we consider another kind of restriction, quite natural when talking about a QPT. We will consider that one (and only one) parameter, say $\lambda$, of the Hamiltonian model is varied to pass through the QPT, keeping other variables or parameters fixed. Thus the metric is unidimensional. We know, that in this case, the geodesic is given by:

$$g_{\lambda\lambda} \, \dot{\lambda}^2 = \mathrm{cte} \tag{17}$$

Therefore,

$$\mathcal{C} = \min_{\lambda(\tau)} \int_0^T \sqrt{g_{\lambda\lambda}} \, \dot{\lambda} \, d\tau \sim T \ . \tag{18}$$

Below, we will work some examples and we will see that $T$ does not diverge at the QPT. However, if we compute the derivative instead:

$$\frac{\partial \mathcal{C}}{\partial \lambda} = \sqrt{g_{\lambda\lambda}} \ . \tag{19}$$

It is known that some components of the metric tensor can diverge, thus diverging the derivative of $\mathcal{C}$. Equation (19) has two consequences. The first one is that, under quite general circumstances, the derivative of $\mathcal{C}$ close to a QPT is related to the metric tensor and inherits its universal properties. Thus, we can utilize the theory of the metric tensor and its behavior near a transition to automatically obtain the scaling for complexity. The second one is that this derivative can be used to witness and characterize QPTs.

### B. Finite size scaling

Here, we review the scaling for the metric tensor [47], from which the behaviour of $\mathcal{C}$ follows directly, Cf. Eq. (19). Close to a critical point correlation length diverges as,

$$\xi \sim |\lambda - \lambda_c|^{-1/a} \ , \tag{20}$$

with $a$ a critical exponent. Similar relations occur for other thermodynamic quantities. In particular, and for what interests us, the metric tensor can be written as [47],

$$g_{\mu\nu} \sim |\lambda - \lambda_c|^{\Delta_{\mu\nu}/a} \ , \tag{21}$$

with $\Delta_{\mu\nu}$ the corresponding critical exponent. Notice that, for the reasons already explained in section I A 3, from now on we will be interested in the intensive part of the metric tensor $g_{\mu\nu} \to g_{\mu\nu}/L^d$, with $d$ the spatial dimensions.

Near a phase transition, finite-size scaling dictates how quantities behave after scale transformations. Very briefly, after a length scale transformation $x' = \alpha x$, time scales as $\tau' = \alpha^z \tau$, with $z$ its critical exponent. This fixes the energy fluctuations $\Delta E \Delta \tau \sim 1 \to \Delta E' = \alpha^{-z} \Delta E$. Putting it all together, it is interesting to extract the value of the critical exponent $\Delta_{\mu\nu}$ above, which controls how the metric tensor behaves, in terms of other critical exponents that dictate more fundamental quantities. Looking at equations (4) and (12) and (13) and writing the scaling for the derivatives of the Hamiltonian as $\partial_{\mu'} \mathcal{H}' = \alpha^{-\Delta_\mu} \partial_\mu \mathcal{H}$ we arrive to [47],

$$\Delta_{\mu\nu} = \Delta_\nu + \Delta_\mu - 2z - d \ . \tag{22}$$

Finally, merging, (20) and (21), we find that close enough to the transition, where the relevant length is given by the system size, $L$, we arrive to

$$g_{\mu\nu} \sim L^{-\Delta_{\mu\nu}} \ . \tag{23}$$

As a consequence of all of this and using (19), when a single parameter is varied across the QPT we have the scaling:

$$\frac{\partial \mathcal{C}}{\partial \lambda} \sim L^{-\Delta_{\lambda\lambda}/2} \ . \tag{24}$$

It is remarkable that the complexity derivative scaling is dictated by universal exponents, whenever one parameter is varied to cross a critical point. In particular, if $\Delta_{\lambda\lambda} > -2$ the derivative is sub-extensive. If $\Delta_{\lambda\lambda} = -2$ it is extensive and if $\Delta_{\lambda\lambda} < -2$ is superextensive.

## III. SOLVABLE HAMILTONIANS

Let us test the above ideas on a couple of solvable models: the one dimensional quantum Ising model [48] and the Dicke [49–51] model.

### A. Quantum Ising model

The transverse field Ising model (Periodic Boundary Conditions will be assumed) is

$$\mathcal{H} = -J \sum_{j=1}^{L} \sigma_j^z \sigma_{j+1}^z + \sum_{j=1}^{L} \sigma_j^x \ . \tag{25}$$

Hamiltonian (25) can be solved via the Jordan-Wigner transformation [48]. This Ising model has a second order phase transition occurring at $J_c = 1(-1)$ in the $L \to \infty$ limit. For $J_c > 1 (J_c < -1)$ the $\mathbb{Z}_2$ symmetry is spontaneously broken and the g.s. is ferromagnetically (antiferromagnetically) ordered. W.l.o.g. we fix our attention in the paramagnetic-ferromagnetic transition occurring at $J_c = 1$. On top of that, the ground state can be written in terms of fermionic excitations (after the Jordan-Wigner transformation) as,

$$|\psi_{gs}\rangle = \prod_{k>0} \left( \cos(\theta_k/2) + i e^{i\phi} \sin(\theta_k/2) \, a_k^\dagger a_{-k}^\dagger \right) |0\rangle \ . \tag{26}$$

with $k = \frac{(2m-1)\pi}{L}$ [3] and,

$$\tan \theta_k = \frac{-J \sin k}{1 + J \cos k} \ . \tag{27}$$

For the rest of the section the metric tensor (12) is needed. It has been computed several times already [52, 53]

$$g_{JJ} = \frac{1}{4} \sum_k \left( \frac{\partial \theta_k}{\partial h} \right)^2 \ . \tag{28}$$

In the thermodynamic limit, the $k$-sum is an integral $\sum_k \to L/\pi \int$ and it can be computed explicitly, yielding

$$g_{JJ} = \frac{-\pi(J^2-1) + i\left(J^2+1\right)\left[\log\left(-\frac{2i(J+1)}{J-1}\right) - \log\left(\frac{2i(J+1)}{J-1}\right)\right]}{32 J^2 (J^2-1)} \ . \tag{29}$$

---

[3] We have used even $L$ and periodic boundary conditions.

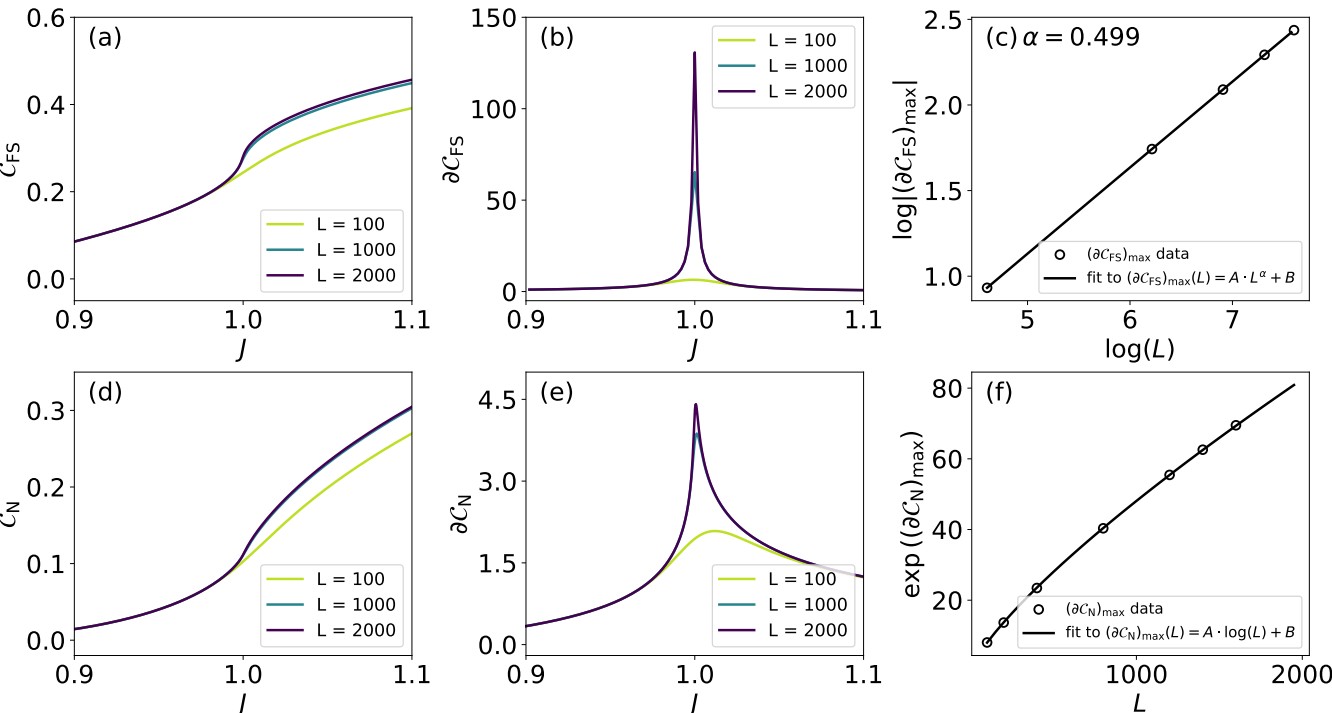

FIG. 1. Study of the complexity for the Transverse Field Ising model. (a) Complexity for different sizes of the chain computed using the Fubini-Study metric. The discretization in $J$ used is $\delta J = 1\mathrm{e}{-3}$. (b) Derivative of the Fubini-Study complexity for different $L$, $\delta J = 2\mathrm{e}{-3}$. (c) Finite size scaling of the maximum in the derivative of the Fubini-Study complexity. See that this maximum diverges polynomially with the size of the chain. (d) Study of the Nielsen complexity, $\delta J = 3\mathrm{e}{-4}$. (e) Derivative of the Nielsen complexity for different $L$, $\delta J = 3\mathrm{e}{-4}$. (f) Finite size scaling of the maximum in the derivative of the Nielsen complexity. See that this maximum diverges logarithmically.

### 1. Complexity through QPTs

From Eq. (29) we see that $g_{JJ}$ diverges at $J = J_c$. This is the reason behind the divergence in the derivative of the complexity at the QPT, Cf. Eq. (19). In figure 1, we plot $\mathcal{C}_{\mathrm{FS}}$ both in the continuum and for $L$-finite using either (29) or the sum (28). In both cases, the integral (18) is computed. It is evident that the complexity does not diverge at the QPT, but its derivative does, inheriting this behaviour from the metric tensor, Cf. Figs. 1a and b. For the Ising transition, the exponent $a = 1$, Cf. Eq. (20). We know that $\Delta_{hh}/a = 1$, so the complexity derivative diverges as $\sim L^{1/2}$ at the Ising transition.

### 2. Relation between $\mathcal{C}_{\mathrm{N}}$ and $\mathcal{C}_{\mathrm{FS}}$

Formula (26) is formally equivalent to the ground state for the 1D-Kitaev model. For the latter, $\mathcal{C}_{\mathrm{N}}$ has been computed in [18]. If the reference, target and intermediate states have the same form (26), $\mathcal{C}_{\mathrm{N}}$ reads:

$$\mathcal{C}_{\mathrm{N}} = \sum_k |\Delta\theta_k|^2 \tag{30}$$

where $\Delta\theta_k = \theta_k^T - \theta_k^R$ and $\theta_k^T$ ($\theta_k^R$) are the angles (27) at the target (reference) states. Following the same procedure as in [18] we checked that $\partial_J C_N \sim \log L$, *i.e.* it diverges logarithmically. This must be confronted with the divergence (with critical exponent $1/2$) for the case of $\partial_J \mathcal{C}_{\mathrm{FS}}$. This is an important difference. While using the FS distance the complexity is associated with the fluctuations, cf. Eq. (13), the $\mathcal{C}_{\mathrm{N}}$ is more related to the angles difference and its divergence is therefore smoothed.

## B. The Dicke model

The Hamiltonian for the ground state sector of the $L$-spin Dicke model can be written in terms of total spin operators of spin $S = L/2$ as [54]

$$H = \omega_c a^\dagger a + \omega_s S^z + \frac{\lambda}{\sqrt{2S}} \left(a^\dagger + a\right)\left(S^+ + S^-\right), \tag{31}$$

where the spin and ladder operators obey the canonical commutation relations $[S^z, S^\pm] = \pm S^\pm$, $[S^+, S^-] = 2S^z$. This model can be solved in the thermodynamic limit, $S \to \infty$, with a Holstein-Primakoff transformation on the spins

$$S^+ \to \sqrt{2S} b^\dagger \sqrt{1 - \frac{b^\dagger b}{2S}}, \tag{32}$$

$$S^- \to \sqrt{2S} \sqrt{1 - \frac{b^\dagger b}{2S}} b, \tag{33}$$

$$S^z \to b^\dagger b - S, \tag{34}$$

$$\tag{35}$$

yielding

$$H = \omega_c a^\dagger a + \omega_s^\dagger a + \lambda \left(a^\dagger + a\right)\left(b^\dagger \sqrt{1 - \frac{b^\dagger b}{2S}} + \sqrt{1 - \frac{b^\dagger b}{2S}} b\right) - \omega_c S. \tag{36}$$

In the normal phase of the Dicke model we can obtain an effective Hamiltonian for $S \to \infty$ by neglecting terms with $2S$ in the denominator in the Hamiltonian of Eq. (36), resulting in a completely symmetric model of coupled harmonic oscillators, one corresponding to the physical oscillator and the other corresponding to the spins within the Holstein-Primakoff transformation

$$H = \omega_c a^\dagger a + \omega_s^\dagger a + \lambda \left(a^\dagger + a\right)\left(b^\dagger + b\right) - \omega_c S. \tag{37}$$

In the superradiant phase, the bosonic modes must be displaced to accommodate the macroscopic occupations that the spins and field develop in this phase. Once the displacements are introduced, terms with powers of $2S$ in the denominator are again neglected in the thermodynamic limit, yielding

$$H = \omega_c \bar{a}^\dagger \bar{a} + \frac{\omega_s}{2\mu}(1+\mu)\bar{b}^\dagger \bar{b} + \frac{\omega_s(1-\mu)(3+\mu)}{8\mu(1+\mu)}\left(\bar{b}^\dagger + \bar{b}\right)^2 + \lambda\mu\sqrt{\frac{2}{1+\mu}}\left(\bar{a}^\dagger + \bar{a}\right)\left(\bar{b}^\dagger + \bar{b}\right), \tag{38}$$

where $\mu = \omega_z \Omega / \left(4\lambda^2\right)$ and $\bar{a}, \bar{b}$ are the displaced bosonic operators [55]. We omit the expressions of the displacement as they are irrelevant in the following. Both the normal and superradiant effective Hamiltonians can be diagonalized in the real space basis, where they present a gaussian profile given by

$$g(x,y) = \left(\frac{\epsilon_+ \epsilon_-}{\pi^2}\right)^{1/4} e^{-\frac{(\mathbf{R}, A\mathbf{R})}{2}}, \tag{39}$$

where $\mathbf{R} = (x,y)$, $x$ and $y$ are the real-space coordinates associated to the modes $a(\bar{a})$ and $b(\bar{b})$, $A = U^{-1}MU$ with $U$ a unitary matrix, $M = \mathrm{diag}\left[\epsilon_-, \epsilon_+\right]$ and $\epsilon_\pm$ are the eigenmodes of the system [56]. The overlap of two different ground states is given by

$$\langle g|g'\rangle = 2\frac{[\det M \det M']^{1/4}}{[\det(M + M')]^{1/2}}. \tag{40}$$

This allows us to compute the components of the quantum metric tensor for the Dicke model exactly in the thermodynamic limit. We combine this with finite size results from exact diagonalization of Hamiltonian (31). The results are shown in Fig. 2. Just like we showed for the case of the Ising model, there is no divergence in $\mathcal{C}_{\mathrm{FS}}$, the only signature of the phase transition is a non-analiticity that is only noticeable in the $L \to \infty$ case. This non-analiticity, or its precursor in the case of finite $L$ is best revealed as a divergence in the derivative of the complexity, which is naturally the square root of the metric tensor. Here we are considering the complexity along a $\lambda$-path and the divergence is

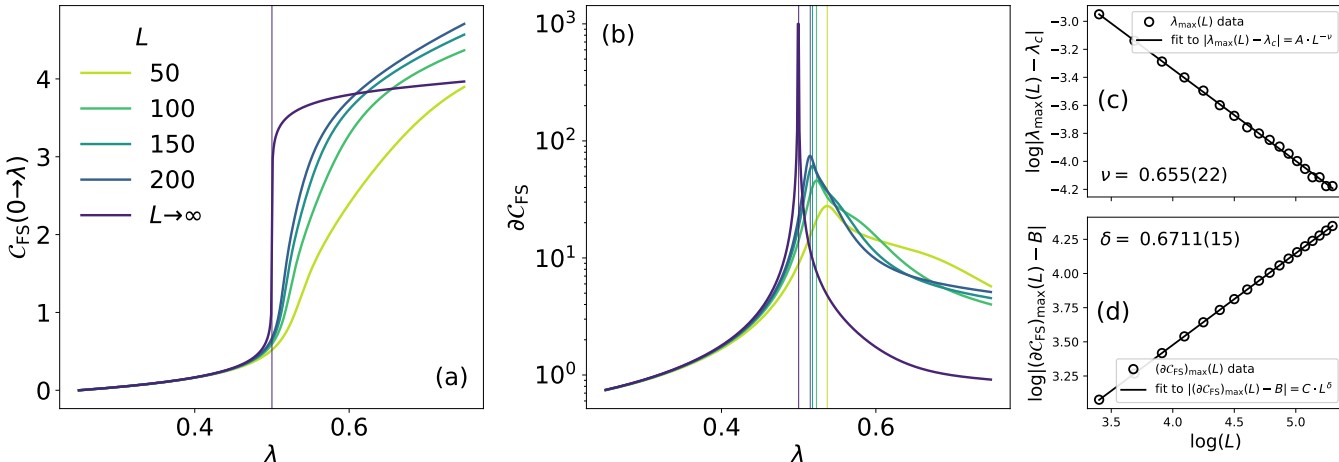

FIG. 2. Fubini-Study complexity (a) and its derivative with respect to $\lambda$ (b) across the phase transition of the Dicke model as a function of the system size (numerical results) and in the thermodynamic limit (analytical results). Plots on the right showcase the fits of $\lambda_{\max}(L)$ (c) and $(\partial \mathcal{C}_{\mathrm{FS}})_{\max}(L)$ (d) (extracted from center plot) to their respective finite size scaling laws. All results are at resonance $\omega_c = \omega_s = 1$ and with a discretization of $d\lambda = 10^{-3}$. Numerical results were obtained with a cutoff for bosonic excitations of $N_{\mathrm{exc}} = 30$ .

revealed in $\partial \mathcal{C}_{\mathrm{FS}} = \sqrt{g_{\lambda\lambda}}$. We perform a finite-size scaling analysis of the metric tensor by fitting the maximal values $(\partial \mathcal{C}_{\mathrm{FS}})_{\max}(L)$ and critical parameters at said maxima $\lambda_{\max}(L)$ to their respective scaling laws

$$|(\partial \mathcal{C}_{\mathrm{FS}})_{\max}(L) - B| = C \cdot L^{\delta} \ , \tag{41}$$

$$|\lambda_{\max}(L) - \lambda_c| = A \cdot L^{-\nu} \ . \tag{42}$$

The resulting critical exponents $\nu = 0.655(22) \cong 2/3$ and $\delta = 0.6711(15) \cong 2/3$ are in agreement with values reported in the literature [57].

## IV. COMPLEXITY IN A QUANTUM COMPUTER, THE CASE OF GROUND STATE PREPARATION

In this section, we compute $\mathcal{C}_{\mathrm{N}}$ when preparing ground states in a quantum computer. We study both adiabatic algorithms and variational quantum eigensolvers (VQEs). Two versions of the former algorithms are discussed: with and without shortcuts to adiabaticity.

In both cases, the initial state is the "trivial zero" $|0\rangle \equiv |00 \cdots 0\rangle$[4]. Some gates are applied to prepare the ground state of a given Hamiltonian. Here, we are especially interested when this initial state (that can be understood as the ground state in the paramagnetic phase) is in a different phase than the final one. In addition, we discuss whether or not a QPT is crossed during the algorithm. Finally, notice that in quantum computing applications it seems natural to compute $\mathcal{C}_{\mathrm{N}}$ and, in particular, its discrete version (the number of gates needed), Cf. Sec. I A 1. Thus, through this section, we compute $\mathcal{C}_{\mathrm{N}}$.

### A. Adiabatic algorithms

A systematic way of finding the ground state of a given Hamiltonian is by adiabatic passage or annealing. Let us consider the time-dependent Hamiltonian:

$$\mathcal{H}(t) = (1 - \lambda(t))\mathcal{H}_0 + \lambda(t)\mathcal{H}_{\mathrm{T}} \ . \tag{43}$$

Here, $\mathcal{H}_0$ has a trivial ground state (easy to prepare), and $\mathcal{H}_{\mathrm{T}}$ is the hamiltonian from which we want to obtain its ground state. Consider that the time-dependent function $\lambda(t)$ runs from $\lambda(t = 0) = 0$ to $\lambda(t = T) = 1$, where $T$ is

---

[4] In fact, in almost all algorithms the initial state seems to be $|00 \cdots 0\rangle$.

the final time of the algorithm. At $t = 0$ the state is prepared in the ground state of $\mathcal{H}_0$. If $\dot{\lambda}$ is sufficiently small compared to the instantaneous gap, by means of the adiabatic theorem the final state is the ground state of $\mathcal{H}_T$ [34]. On the other hand, the adiabatic condition alerts us that as the gap closes, for example in continuous phase transitions, the execution time, *i.e.* the circuit depth, scales with the inverse of this gap, thus also $\mathcal{C}$.

Importantly enough the adiabatic condition can be relaxed by introducing counter-diabatic terms. Generally speaking, instead of $\mathcal{H}(\tau)$ (whose ground states are $|\psi(t_n)\rangle$) what is evolved is the "modified" Hamiltonian [58, 59]:

$$\mathcal{H}'(\tau) = \mathcal{H}(\tau) + \mathcal{H}_{\text{CD}}(\tau) \tag{44}$$

The last term ensures that the time evolution exactly matches the instantaneous ground state of $\mathcal{H}(\tau)$ no matter how fast the evolution is. This is known in the literature as shortcuts to adiabaticity and $\mathcal{H}_{\text{CD}}$ is called counter-diabatic Hamiltonian. There are different ways of writing $\mathcal{H}_{\text{CD}}$. In its original form we can write:

$$\mathcal{H}_{\text{CD}}(\tau) = i\dot{\lambda}^{\mu} \sum \frac{\langle m|\partial_{\mu}\mathcal{H}|n\rangle}{E_n - E_m}|m\rangle\langle n| \; + \text{h.c.} \tag{45}$$

with $\partial_{\mu}\mathcal{H} \equiv \partial\mathcal{H}/\partial\lambda^{\mu}$. We emphasize that at times 0 and $t$, $|\psi_R\rangle$ and $|\psi_T\rangle$ are ground states of $\mathcal{H}(0)$ and $\mathcal{H}(t)$ respectively. Explicitly $|\psi_T\rangle = \mathcal{T}e^{-\int_0^t \mathcal{H}'(\tau)\,d\tau}|\psi_R\rangle$. Here $\tau$ means time, cf. with Eq (1). To connect this evolution with the previous sections, we note that the fidelity susceptibility can be written in terms of the $\mathcal{H}_{\text{CD}}(\tau)$ fluctuations [Cf. Eq. (9) and (13)]:

$$\chi_F = \langle(\mathcal{H}(\tau) + \mathcal{H}_{\text{CD}}(\tau))^2\rangle - \langle(\mathcal{H}(\tau) + \mathcal{H}_{\text{CD}}(\tau))\rangle^2 = \langle\mathcal{H}_{\text{CD}}(\tau)^2\rangle = \dot{\lambda}^{\mu}\dot{\lambda}^{\nu}\,g_{\mu\nu}\;. \tag{46}$$

In practice $\mathcal{H}_{\text{CD}}$ is difficult to find. Therefore, a systematic although approximate way of writing is convenient. Following [60] it can be rewritten as,

$$\mathcal{H}_{\text{CD}}(\tau) = \dot{\lambda}^{\mu}\mathcal{A}_{\mu} \tag{47}$$

Here, $\mathcal{A}$ is the adiabatic gauge potential that can be approximated as:

$$\mathcal{A}_{\mu}^{(\ell)} = i\sum_{k=1}^{\ell}\alpha_k\underbrace{[\mathcal{H},[\mathcal{H},\dots[\mathcal{H},\partial_{\mu}\mathcal{H}]]]}_{2k-1} \tag{48}$$

where $(l)$ is the "degree of approximation". On top of that, the $\{\alpha_k\}$ are found variationally by minimising the action [61]:

$$S_{\ell} = \text{Tr}\left[G_{\ell}^2\right], \quad G_{\ell} = \dot{\lambda}^{\mu}\left(\partial_{\mu}\mathcal{H} - i\left[\mathcal{H},\mathcal{A}_{\mu}^{(\ell)}\right]\right) \tag{49}$$

In many cases of interest, in the adiabatic protocol, $\mathcal{H}(\tau)$ is expected to be a local Hamiltonian, in particular a two body one. Notice that due to nested commutators, the higher the order $(l)$, the longer the range of interaction. Following the functional (6) three, four, or higher order body interactions will be highly penalised. Thus, in what follows, we will restrict ourselves to $l = 1$ that introduces two body interactions at most. This, in turn, provides a systematic way of preparing, via trotterization, quantum states.

### 1. Complexity in adiabatic algorithms

As has been previously discussed, in order to compute the complexity as defined by Nielsen [Cf. Sec.(I A 1)], we only need to express our unitary operation as the time evolution of some Hamiltonian. In the present case, it is straightforward, with and without shortcuts, as the Hamiltonian (44) is given explicitly. We study the Ising model in transverse field and the ZZXZ model. For both, we use the function $\lambda(t) = \sin^2\left[\frac{\pi}{2}\sin^2\left(\frac{\pi t}{2T}\right)\right]$ to drive the evolution from the initial Hamiltonian, $\mathcal{H}_0$, to the target one, $\mathcal{H}_T$.

For the Ising model, we start from $\mathcal{H}_0 = h_x\sum_i\sigma_i^x$, leaving the transverse field fixed at value $h_x = 1$ and switching on the spin-spin interaction until we reach $\mathcal{H}_{\text{int}} = J\sum_i\sigma_i^z\sigma_{i+1}^z$. The counter-diabatic Hamiltonian follows from equations (45), (48) and (49) with $l = 1$ yielding $\mathcal{H}_{\text{CD}}(t) = \dot{\lambda}(t)\alpha(t)\sum_i(\sigma_i^y\sigma_{i+1}^z + \sigma_i^z\sigma_{i+1}^y)$, where $\alpha(t)$ is the variational parameter in Eqs. (48) and (49). The full-time-dependent Hamiltonian reads

$$\mathcal{H}'(t) = \mathcal{H}_0 + \lambda(t)\mathcal{H}_{\text{int}} + \mathcal{H}_{\text{CD}}(t)\;. \tag{50}$$

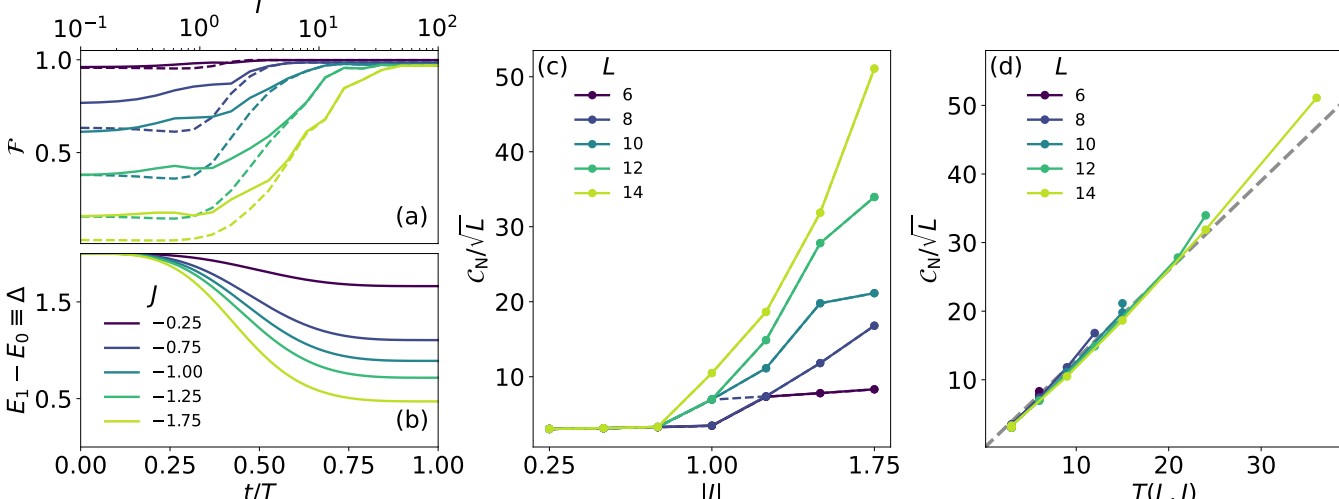

FIG. 3. Complexity study for the Transverse Field Ising model using the adiabatic algorithm. (a) Evolution of the fidelity obtained with shortcuts to adiabaticity (solid lines) and without them (dashed lines) for increasing time lengths of the full algorithm and different target $J$s for $L = 12$. At shorter times the shortcuts provide better results, being identical to the simple case (without shortcuts) for the longest times. (b) Evolution of the gap between the ground state and the first excited state during the algorithm for the same values of $J$ as in (a). The gap closes with an increasing value of $|J|$, explaining why longer times are needed for the larger $|J|$ to obtain the same fidelity. (c) Complexity computed for different sizes with shortcuts (solid lines) and without shortcuts (dashed). As the gap closes, more gates are needed to achieve the fidelity threshold (0.9 in this case) but we do not find relevant differences between applying shortcuts or not in the final result for the complexity. (d) Complexity dependence on the chain size (root squared) and the preparation time. Each point corresponds to the preparation time for each $|J|$ in (c).

Before discussing the complexity in adiabatic algorithms, let us investigate how much time is needed to reach the ground state and its relation to the Hamiltonian gap. It is well known that $T$ determines the practicality of the algorithm, and in this section, we will explore its relationship with $\mathcal{C}_N$. To implement the unitary evolution via Trotter decomposition, we need to split the total time $T$ into $T/\delta T$ steps, where $\delta T$ is the time discretization employed. A smaller $\delta T$ ensures a more precise implementation but requires more gates, increasing the computational cost of the implementation. In the simulations presented here, we use $\delta T = \min(0.1, T/30)$.

In Figure 3a, we plot the fidelity between the final state obtained adiabatically and the target state. As expected, longer times result in better fidelity. Additionally, we confirm that at lower times, higher fidelities are achieved thanks to the counter-diabatic term. Figure 3b shows the gap evolution within the adiabatic algorithm. As $|J|$ increases and the gap between the first excited state and the ground state closes, more time is required for the preparation.

Following Eq. (6), the Nielsen complexity $\mathcal{C}_N$ for the adiabatic algorithm is given by:

$$\mathcal{C}_N = \int_0^T dt \left[ L + (L-1)\lambda(t)^2 J^2 + 2(L-1)\dot{\lambda}^2(t)\alpha^2(t) \right]^{1/2} . \tag{51}$$

Here, we set the final time $T$ as the time required for the adiabatic algorithm to reach a certain fidelity threshold, which in our case will be $\mathcal{F} = 0.9$. From the above formula, we can extract a $\sqrt{L}$. Figure 3c shows the actual Nielsen complexity values. Reflecting the fidelity behavior, the complexity jumps around the transition as the gap is closing. From Eq. (51), $\mathcal{C}_N$ is proportional to $T$, and $T$ depends on the gap, so this jump in complexity is expected when approaching the critical region.

More interesting is what we show in panel 3d, where $\mathcal{C}_N/\sqrt{L}$ shows universal behavior independent of $L$ when plotting it as a function of $T$. Additionally, we observe that $\mathcal{C}_N \sim T$. This can be understood as follows. First, we can ignore the case with shortcuts, as it barely affects the complexity (see Figure 3b). By doing so, Eq. (51) is simplified as follows:

$$C_N \cong T\sqrt{L} \int_0^1 dx \sqrt{1 + J^2\lambda^2(x)} = T\sqrt{L} \times \begin{cases} 1 + \frac{1}{16}(3 - J_0(\pi))J^2 \cong 1 + 0.2J^2 & J \ll 1 \\ J/2 & J \gg 1 \end{cases} \tag{52}$$

Here, $J_0(\cdot)$ is the Bessel function. In the $J$-range studied here, the first approximation for the integral is quite accurate. This introduces a dependence on $J^2$, however, this dependence is still small, and the overall dependence is very well

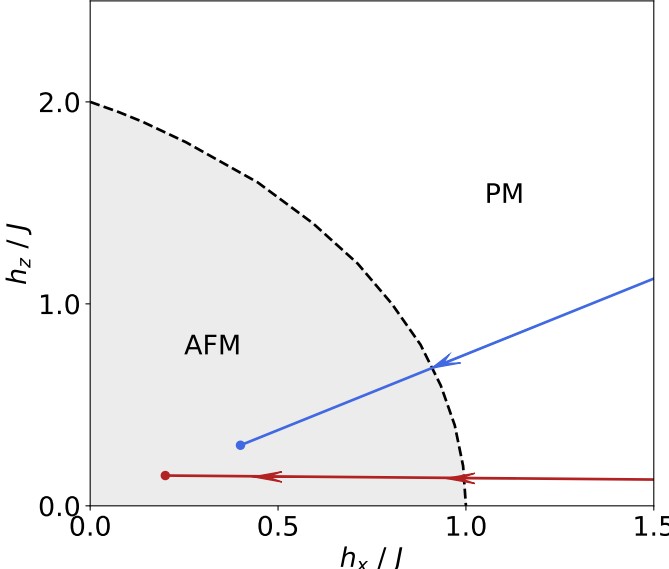

FIG. 4. Phase diagram of the ZZXZ model for zero temperature. The black dotted line signals the critical region between phases for different ratios of the fields $(h_x, h_z)$ to the magnitude of the exchange interaction $(J)$. The coloured lines depict the path followed for the adiabatic algorithm (red) and the values computed in the VQE (blue) [Cf. Sec. IV B].

approximated by $\mathcal{C}_N \propto T\sqrt{L}$, as confirmed by our numerical results (dashed line in Figure 3 d). As a consequence, we can use some results from the adiabatic theorem relating the final $T$ and the minimum gap in Eq. (55). It has been proven that, in the best case, the preparation time scales with the minimum energy gap as $T \sim \Delta^{-2}$ [34], with $\Delta \equiv \min_{j \neq k} |E_j - E_k|$. This dependency is inherited by the complexity and is shown in Appendix C. There, we verify that $\Delta^{-2}$ holds as long as the gap is not too small, where the dependency is lost. This may be attributed to several reasons, such as the precision of our numerics or the fact that our 0.9-fidelity threshold cannot discern in those cases.

In order to check if this holds in other models, we also study the so-called antiferromagnetic ZZXZ model:

$$\mathcal{H}_T = J \sum_i \sigma_i^z \sigma_{i+1}^z + h_x \sum_i \sigma_i^x + h_z \sum_i \sigma_i^z . \tag{53}$$

Due to the combination of longitudinal and transverse fields, this is a non-integrable model. It is ideal, then, to explore the phenomenology of complexity beyond the exactly solvable models considered so far. In Fig. 4 we draw the phase diagram of the model at zero temperature as a function of the fields applied to the spins and the exchange constant [62]. The critical line separates paramagnetic and antiferromagnetic phases. For our particular purposes, keeping the same initial Hamiltonian, $\mathcal{H}_0$, we set the transverse field, $h_x = 1$ and the target longitudinal field to $h_z = 0.75$. We thus study the quantum phase transition appearing when moving to different target values of $J$. This path is shown as the red line in Fig. 4, where the final point marks the maximum value simulated for the target $J$. Therefore, the transverse field is going to be fixed while we turn on both the longitudinal field and the magnetic interaction. The counter-diabatic term can be computed in the same fashion as before, getting the same result as in [60]. The time-dependent Hamiltonian reads

$$\mathcal{H}'(t) = \mathcal{H}_0 + \lambda(t) \sum_i \left( J\sigma_i^z \sigma_{i+1}^z + h_z \sigma_i^z \right) + \mathcal{H}_{CD}(t) \tag{54}$$

and the complexity acquires the following expression

$$C_N = \int_0^T dt \left[ L \left( 1 + h_z^2 \lambda^2(t) \right) + (L-1)\lambda(t)^2 J^2 + \dot{\lambda}^2(t) \left( L\alpha^2(t) + 2(L-1) \left( \beta^2(t) + \gamma^2(t) \right) \right) \right]^{1/2} . \tag{55}$$

In figure 5 we show the results obtained for the different values of $J$ and the chain sizes, $L$. The behaviour is equivalent to the previous model except that for sufficiently large values of $J$, the gap decreases sharply, closing completely (see figure 5b), causing the counter-diabatic terms to cause more error than the simple evolution itself, as we can see in panel (a) of the same figure. This is a consequence of the fact that our expression for the counter-diabatic term is

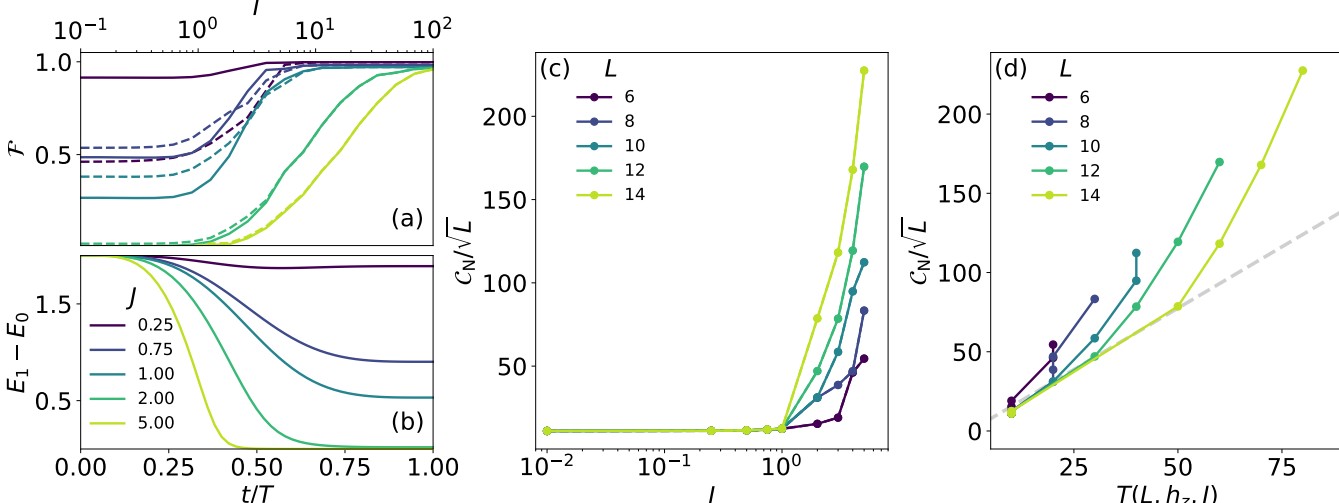

FIG. 5. Complexity study for the ZZXZ model using the adiabatic algorithm. The phenomenology is essentially the same as for the TFI model. (a) Evolution of the fidelity obtained with shortcuts to adiabaticity (solid lines) and without them (dashed lines) for increasing time lengths of the full algorithm and $L = 12$. In this case we see that, for sufficiently large values of $J$, no applying shortcuts works better than applying them. This is explained by the gap closing much more abruptly than in the TFI model, as can be seen in (b). (c) Normalised complexity computed for different sizes with shortcuts (solid lines) and without shortcuts (dashed). As the gap closes, more gates are needed to achieve the fidelity threshold (0.9 in this case). (d) Complexity dependence on the chain size (root squared) and the preparation time. Each point corresponds to the preparation time for each $|J|$ in (c).

not exact, but a first-order approximation of a general expression [cf Eq. (48)]. This scenario serves to illustrate that the design of shortcuts can be tricky. Solutions to this problem could be to go to higher orders in the development of the CD term or to explore other $\lambda(t)$ functions as in [63, 64], where geometric arguments are used to obtain the optimal way to vary the time dependent parameters in the adiabatic evolution.

We can repeat the study in Eq. (52) (for the dependencies of the complexity on the system size, preparation time and model parameters) for this model. As one can see in Eq. (56), the only change is in the integral.

$$\mathcal{C}_\mathrm{N} \cong T\sqrt{L} \int_0^1 dx \sqrt{1 + (h_z^2 + J^2)\lambda^2(x)} = T\sqrt{L} \int_0^1 dx \sqrt{1 + \tilde{J}^2 \lambda^2(x)} \tag{56}$$

Despite obtaining an analogous analytical expression, we do not recover a linear dependence of $\mathcal{C}_\mathrm{N}/\sqrt{L}$ on $T$ in practice, as shown in Fig. 5d. Two effects contribute to the deviation. First, since we are now reaching larger values of $J$, the integral becomes more relevant than in the previous model. Thus, the complexity depends on $J$ in two ways: through the preparation time $T$ and through the integral. As a result, the scaling of the complexity with $T$ is now superlinear, as a change in $T$ is influenced by an underlying change in $J$ which also contributes to the complexity through the integral. The second effect is numerical. The preparation time for this model is longer than for the TFI model, as shown in the other panels of Fig. 5. As a consequence, our discretization in preparation time is now insufficient to resolve the true dependence of $\mathcal{C}/\sqrt{L}$ on $T$. This effect manifests as bunching: we observe several different values of complexity for the same value of $T$. This is because the complexity is changing with $J$ through the integral but the preparation time is stuck at the closest larger value. This effect should disappear with finer numerics.

The inclusion of shortcuts does not provide any significant advantage in terms of complexity reduction. This is because we have constrained these shortcuts to be as local as possible, in our case $l = 1$ in (49), introducing two body interactions at much. It is expected that by introducing long-range terms in (45) the complexity decreases as the system approaches to the QPT. This can be compared to the previous section II, where there was no restriction to local operations, so both $\mathcal{C}_F$ and $\mathcal{C}_\mathrm{N}$ remained finite despite crossing the QPT. Other paths investigated in this work are sent to App. B.

## B. Circuit Complexity in VQEs

VQEs, introduced in [36], use the fact that any quantum state can be written in terms of a unitary operation as

$$|\phi(\vec{\theta})\rangle = U(\vec{\theta})|0\rangle \ , \tag{57}$$

where $U(\vec{\theta})$ is a parameterized unitary that transforms the initial state into the desired wave function $|\phi(\vec{\theta})\rangle$. This unitary can be implemented in a quantum circuit as a set of quantum gates. The expectation value of the Hamiltonian where we encode our problem ($\mathcal{H}$) results

$$\langle\mathcal{H}\rangle = \langle 0|U^{\dagger}(\vec{\theta})\mathcal{H}U(\vec{\theta})|0\rangle \geq E_0 \ . \tag{58}$$

The optimization process consists on minimizing the average energy of the parameterized state:

$$E_{\text{VQE}} = \min_{\theta} \langle 0|U(\vec{\theta})^{\dagger}\mathcal{H}U(\vec{\theta})|0\rangle \geq E_0 \ . \tag{59}$$

The algorithm can be divided into three different stages. First, we need to choose the trial wave function (see Eq.(57)). Choosing the unitary $U(\vec{\theta})$ is equivalent to constructing the quantum circuit that transforms the initial state into the parameterized wave function. The circuit used to achieve $|\phi(\vec{\theta})\rangle$ is called the *ansatz* and can be represented as,

$$\tag{60}$$

Choosing an appropriate ansatz is crucial for the optimization process. This choice depends completely on the model we are simulating and the set of gates available. We will dig into our choice of unitary below. The next step is constructing the Hamiltonian of the problem. Since this Hamiltonian is going to be evaluated later, Eq. (58), it must be written in terms of Pauli strings $\{\mathbb{I}, \sigma_x, \sigma_y, \sigma_z\}^{\otimes L}$. Pauli operators are related to spin observables, which are suitable for direct measurement in quantum devices [65]. With the Hamiltonian and the wave function defined, we can measure the energy of the state, which is the cost function. To compute this cost function, the expectation values of the Pauli observables are measured determining the value of the energy. Since the technique uses quantum and classical processors, VQEs are cast as hybrid algorithms. Our results are numerical and our Python code simply computes the product of the matrices $U(\vec{\theta})^{\dagger}\mathcal{H}U(\vec{\theta})$ previously defined and then projects onto the zero state obtaining $\langle 0|U(\vec{\theta})^{\dagger}\mathcal{H}U(\vec{\theta})|0\rangle$. We will not discuss its measurement overhead. Here, we are interested in the *circuit complexity for reaching the desired ground state*.

The final step is to minimize this cost function through the variation of the parameters $\theta$ in the wave function. At the end of each iteration we obtain the value of the energy (59). Then, a classical optimizer determines the best direction of variation of the parameter vector $\vec{\theta}$ to minimize this value. We use as many iterations as needed until we converge to a final solution for the coordinates of the parameter vector. Ideally, this solution is the absolute minimum in the space of parameters. Still, obtaining this minimum is not an easy task. The optimizer can get trapped in local minima which will imply serious limitations in the minimization process. This problem and others have been previously discussed in the literature [65, 66] and are out of scope for this work.

Summarizing, we assume a given ansatz, the set of available gates in $U(\vec{\theta})$ in (57) and the hybrid algorithm finds the optimal solution. $\mathcal{C}_{\text{N}}$ counts the number of gates, and once the VQE circuit is chosen, it can be done systematically.

### 1. Local VQE ansatz

We focus on a fixed geometry that is suitable for one-dimensional systems with single and two-qubit gates, besides the two-qubit gates act only on contiguous qubits. This ansatz can be interpreted as a Trotter approximation of

continuous evolution by a local 1D Hamiltonian [67]. In this case, we can separate the terms of the Hamiltonian that act on even and odd links and obtain two sets, each made of mutually commuting gates. In particular, the circuit is given by

$$(61)$$

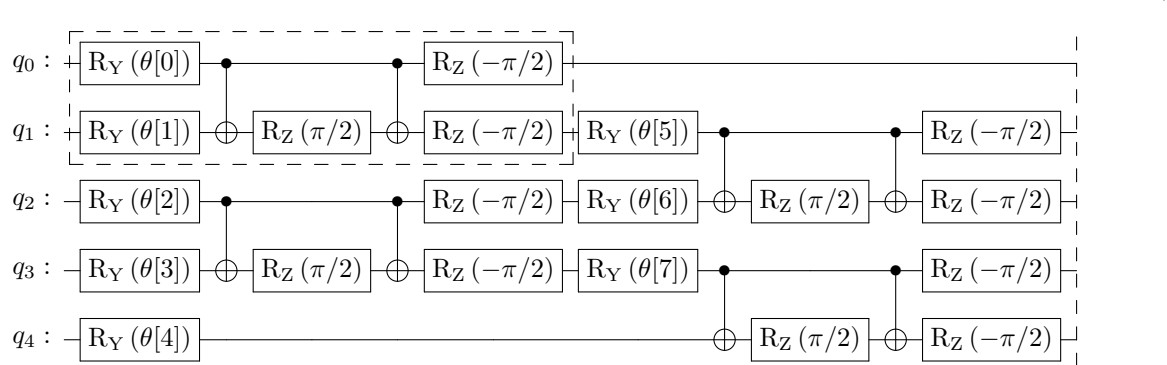

*i.e.* it consists of fundamental blocks (or layers) (separated by dashed lines above). Each layer is made out of single-qubit rotations $R_y(\theta)$ and control-Z gates (CZ). At the end of the circuit, we add a final column of rotations $(R_y)$.

For computing $\mathcal{C}_N$ we rewrite the CZ gates in terms of Pauli operators, count the gates and use equation (6). This is a routine process that we send to Appendix A. Here, we just give the final result:

$$\mathcal{C}_N = \sum_{j=1}^{d} \sqrt{\sum_{i}^{2(L-1)} \left(\frac{\theta_i^j}{2}\right)^2 + 3(L-1)\left(\frac{\pi}{4}\right)^2} \; . \tag{62}$$

### 2. VQE complexity through QPTs

As before, we focus on Ising and ZZXZ models, Eqs. (25) and (53). In Figure 6 we summarize our results for the Ising Hamiltonian. In panels a-c) we plot the complexity using the local VQE ansatz to obtain the ground state at a given $J$ for different chain sizes. We see that $\mathcal{C}_N$ grows when the ground state approaches the QPT, that in this case is given by $J_c \cong 1$ [5]. In fact, close enough to the transition, the VQE cannot reach an acceptable ground state for a maximum depth of 8 (in our simulations). This can be checked in panel d) where the fidelity between the state obtained within the VQE algorithm and the exact ground state falls below 0.9 in the gray region of panel a) for $L = 12$.

It is possible to understand why the VQE fails around the QPT (gray zones in Figure 6). The local ansatz in Eq. (61) is a trotter-like decomposition of a time-dependent two-body interaction Hamiltonian [see Appendix A]. As a result, the local *ansatz* can generate states whose correlation length, $\xi$, grows linearly with the number of layers (circuit depth), as described in detail in [67], which is rooted in Lieb-Robinson bounds. Thus, we have:

$$\xi(d) \sim d \; .$$

On the other hand, for finite systems close to the phase transition, the correlation length saturates, $\xi \to L$. Therefore, the lower bound for the circuit depth is $L$. Additionally, from the calculated VQE-complexity, Eq. (62), it scales as,

$$\mathcal{C}_N \sim \sqrt{L} \times d, \tag{63}$$

Consequently, close to the QPT,

$$\mathcal{C}_N \gtrsim L^{3/2} \; . \tag{64}$$

This bound applies to the local ansatz and is specific to 1D systems. Our numerical simulations become intractable for $d \geq 8$, so for reasonable sizes the VQE ansatz cannot approximate the ground state, explaining the gray zones.

---

[5] We say $J_c \cong 1$ since our simulations are done in finite systems. $J_c = 1$ in the thermodynamic limit.

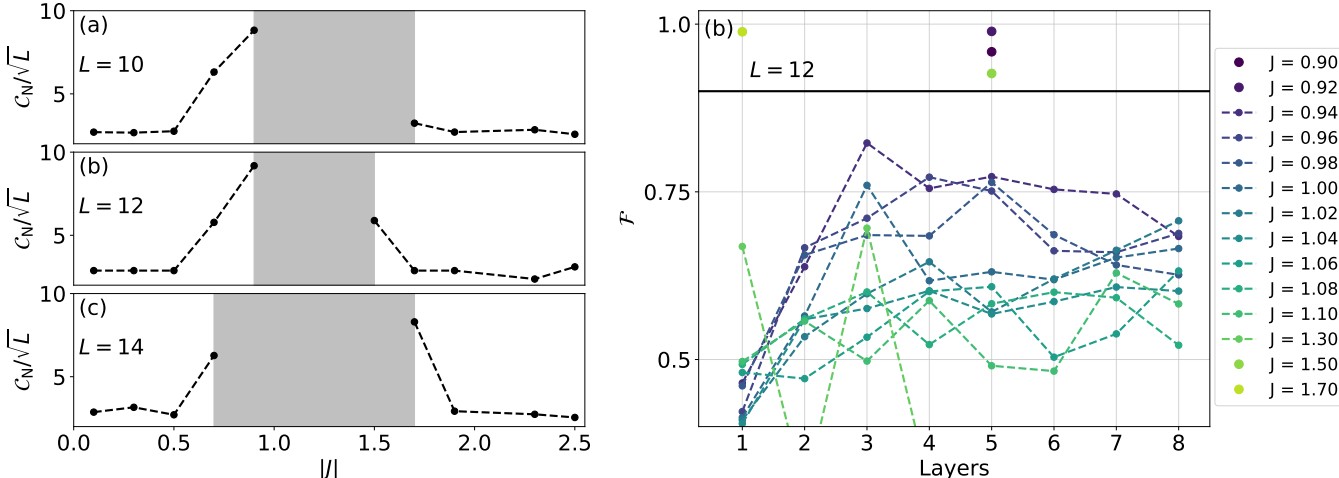

FIG. 6. Transverse Field Ising model with bias, $\epsilon = 0.001$. (a-c) Complexity as a function of $J$ for sizes $L = 10, 12, 14$. The grey zone indicates that the VQE does not converge for points inside that region in a reasonable number of layers to the fidelity threshold (0.9). (d) Fidelity obtained for different numbers of layers for points inside the grey box in (a) and in its vicinity for $L = 12$. For those points whose fidelity is above the threshold (0.9) only the best result has been plotted, for clarity's sake.

One final comment. Non-local *ansatzes* are expected to reduce the circuit depth, as discussed in [68]. Further research on the complexity in this context would be interesting. It is worth noting that in the original proposal, long-range interactions are penalized in the functional $F(\tau)$, see Eq. (6).

Now, let us explain what happens when the target state is far away from the quantum phase transition (QPT). Before delving into the specifics, it is important to note that in finite simulations, deep within the ferromagnetic phase, the $\mathbb{Z}_2$ symmetry is not broken. Therefore, the ground state manifold found by exact diagonalization is spanned by the states $\frac{1}{2}(|0,...,0\rangle \pm |1,...,1\rangle)$. However, through energy minimization, the VQE reaches one of the fully polarized states, either $|0,...,0\rangle$ or $|1,...,1\rangle$, given that they are degenerate with the symmetric ground state. Our convergence criterion is based on reaching a fidelity of 0.9 between the state generated by the VQE and the result of exact diagonalization. Due to the discrepancy in the ground states obtained by both methods in the ferromagnetic phase, the fidelity is capped at 0.5, and the convergence criterion is never satisfied. To address this discrepancy and align with the physics of actual QPTs in the thermodynamic limit, where the symmetry is (spontaneously) broken, we decide to introduce a small bias, $\epsilon \sum \sigma_i^z$, in (25).

That being said, it is remarkable that when the target state is far from the critical point, the complexity decreases, even though the target point and the reference state may belong to different phases. This is because, unlike the adiabatic algorithm, the VQE does not necessarily need to traverse states in the transition region to transition from $|\psi_R\rangle$ to $|\psi_T\rangle$; it can bypass criticality. This phenomenon is easy to understand in the Ising model since, in the paramagnetic phase, the ground state is approximately given by $|+,...,+\rangle$ ($|+\rangle = 1/\sqrt{2}(|0\rangle + |1\rangle)$), as shown in Eq. (25). This state is straightforward to prepare since it can be obtained through single-qubit rotations from the reference state $|\psi_R\rangle = |0,...,0\rangle$. On the other hand, in the ferromagnetic phase, with the added bias, the ground state is either $|0,...,0\rangle$ or $|1,...,1\rangle$, which can also be easily obtained within the VQE.

We now consider the ZZXZ model, Hamiltonian (53)[6]. Here, we are not going to explicitly break the symmetry in order to discuss the scenario in which the symmetric ground state is sought. In the ZZXZ model, the QPT separates paramagnetic (PM) and antiferromagnetic (AFM) phases. In the PM phase, the behavior is analogous to the Ising model, Cf. Figs. 6 and 7. Deep in the AFM phase, the ground state manifold is spanned by the states $|\psi_{\text{AFM}}\rangle \cong \frac{1}{\sqrt{2}}(|1,0,1,0,...\rangle \pm |0,1,0,1,...\rangle)$. Following the previous discussion, the VQE does not reach the symmetric ground state. Therefore, we see that $\mathcal{C}_N$ grows as it approaches the phase transition (with our parameters $J_c \lesssim 1$, see Fig. 4) but does not decrease afterwards. At some point near criticality, the VQE cannot produce a ground state with a fidelity larger than 0.9, see panel d) for the $L = 12$ case, similar to the TFI model scenario. Here, however, the state remains difficult for the VQE after the near-transition region is surpassed. This is further confirmed in figure 8. There, we can see that although the total magnetization is well reproduced by the VQE (also the energy, in panel c), once we enter the antiferromagnetic phase the VQE generates either $|1,0,1,0,...\rangle$ or $|0,1,0,1,...\rangle$, as can be seen

---

[6] The parameters employed in the simulations are depicted as the blue line in Fig. 4, namely $h_x = 1$, $h_z = 0.75$ and $J \in (0., 2.5]$

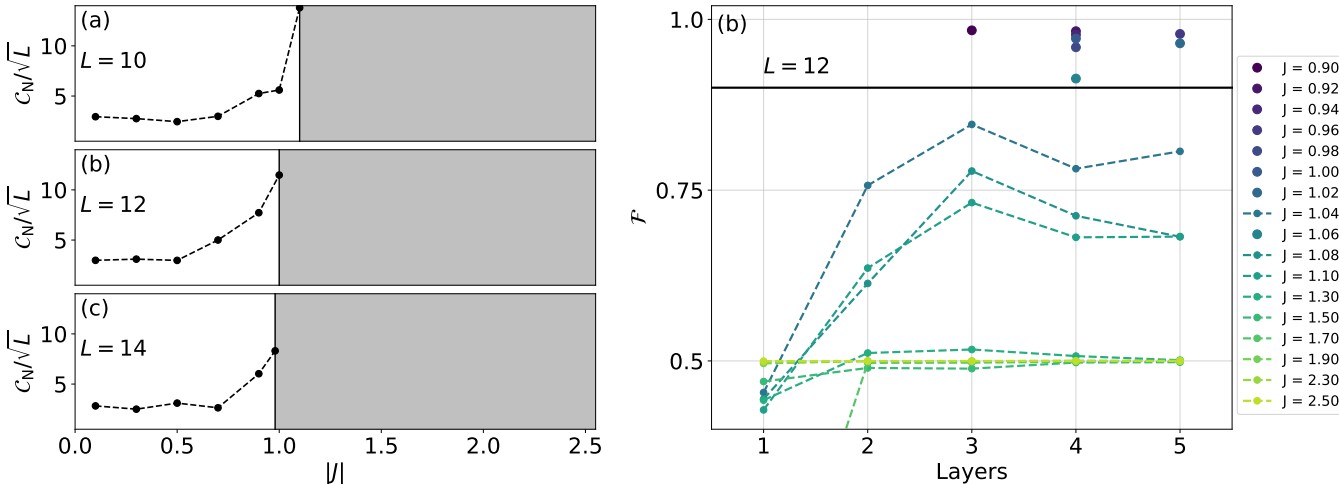

FIG. 7. ZZXZ Ising model. (a-c) Complexity as a function of $J$ for sizes $L = 10, 12, 14$. The grey zone, as in the TFI model, indicates that the algorithm fails to achieve fidelity over 0.9 for points within that region. (d) The fidelity behaviour with the depth of the *ansatz* shows that, again, once the QPT is crossed the algorithm cannot reach fidelities over 0.9. In contrast to the TFI model, here we don't recover high fidelity once we are fully in the antiferromagnetic phase, reaching a maximum value of 0.5 for the highest values of $J$ ($L = 12$).

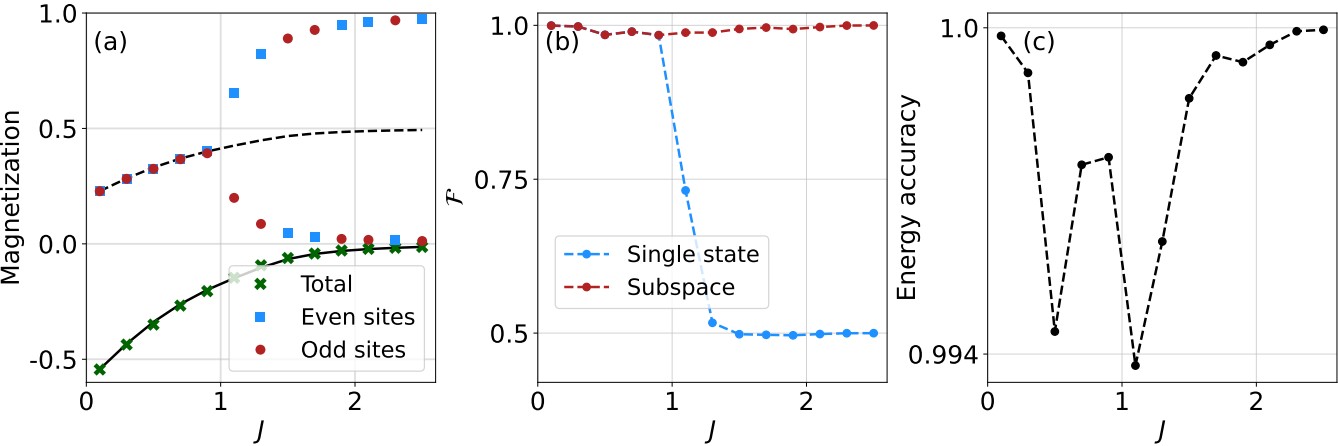

FIG. 8. VQE state characterization in the ZZXZ model for $L = 12$. (a) Magnetization of the spin chain as a function of $J$ obtained from the states generated by VQE. The solid black line represents the total magnetization per site that the spin chain should have (obtained via exact diagonalization) whereas the dashed black line sets the magnetization per site in even/odd sites. (b) Evolution of the best fidelity obtained as a function of $J$. In blue it is computed the fidelity as the overlap between the state generated by the VQE and the exact ground state; in red it is computed as the projection onto the subspace generated by the ground state and the first excited state. (c) Energy accuracy obtained for the same configurations displayed in the other panels computed as $1 - E_{\rm rel}$, being $E_{\rm rel}$ the relative error between the energy obtained from VQE and the exact value.

by computing the magnetization per site, which should be close to $1/2$ in the exact ground state. However, the VQE gives 0 (1) for the even (odd) sites. To conclude our characterization, we see that all this is consistent with obtaining a $\mathcal{F} = 0.5$, as well as a $\mathcal{F} \cong 1$ if we compare the state generated by the VQE with the projection onto the subspace generated by the ground state and the first excited state.

## V. DISCUSSION

Knowing in advance how much a computation will cost, even if only approximately, is of great help. Unfortunately, this estimation can pose a great challenge. Computer science has traditionally categorized problems into different complexity classes, allowing one to know whether a given problem is tractable on a *classical* computer. For a

quantum computer, we can ask a similar question to know if the task we want to tackle is going to be feasible with the architecture we have at hand. For this purpose, the concept of circuit complexity was *invented*. Again, knowing the complexity of each task in any architecture seems too general to be able to give a concrete answer. On the other hand, we can shed some light on generic situations where some kind of general statement can be made. This is the idea that motivated us to write this manuscript. We have studied the situation in which a critical region is crossed in the process of preparing a state.

Our work has shown that, regardless of the type of complexity one chooses, and for diverse models, it appears that complexity grows if the algorithm visits states near a phase transition. We have further proven that this is a characteristic trait of typical algorithms for state preparation such as VQE and adiabatic evolution. The degree of divergence does depend on the definition of complexity used and on the allowed gates. In the case of local *ansätze* or evolutions, $\mathcal{C}$ tends to diverge as the system size grows, specifically as $\sim L^{3/2}$. Importantly, we have shown that VQEs, to the extent that they can go "directly" from the reference to the target state, can potentially avoid the divergence in complexity even if the reference and target states lie in different classes. Whether this is possible depends on the model, as it is determined by the degree of entanglement of the target and reference states. In the case of adiabatic algorithms, we have shown that complexity is bounded by the $T$, wich represents the total time of the algorithm. Therefore, for keeping the complexity down seems to be a matter of allowing non-local gates in the evolution, to fully exploit shortcuts to adiabaticity. This is supported analytically in Sec. III. Here, the Ising critical point is traversed along a restricted path of states of the form (26). Despite this restriction, these states are sufficiently non-local for $\mathcal{C}_N$ to remain finite.

The impact of our work on the preparation of states in a quantum machine seems straightforward. What our results mean in the field of holography is another matter. Unfortunately, we do not have the knowledge to anticipate anything, but it would be interesting to think in this direction. Other ideas not discussed here would be the use of other types of complexity such as Krylov [23, 69–72] or mixed states and their behavior in thermal phase transitions. We leave this for future work.

*Note Added in Proof.-* While we were finishing writing this manuscript, the paper [73], which discusses the importance of local and non-local gates in the computation of complexity, appeared in the arXiv.

## ACKNOWLEDGMENTS

The authors thank Fernando Luis for his helpful comments and insights during the preparation of this manuscript. The authors acknowledge funding from the EU (QUANTERA SUMO and FET-OPEN Grant 862893 FATMOLS), the Spanish Government Grants PID2020-115221GB-C41/AEI/10.13039/501100011033 and TED2021-131447B-C21 funded by MCIN/AEI/10.13039/501100011033 and the EU "NextGenerationEU"/PRTR, the Gobierno de Aragón (Grant E09-17R Q-MAD) and the CSIC Quantum Technologies Platform PTI-001. This work has been financially supported by the Ministry of Economic Affairs and Digital Transformation of the Spanish Government through the QUANTUM ENIA project call - Quantum Spain project, and by the European Union through the Recovery, Transformation and Resilience Plan - NextGenerationEU within the framework of the Digital Spain 2026 Agenda". J R-R acknowledges support from the Ministry of Universities of the Spanish Government through the grant FPU2020-07231. S. R-J. acknowledges financial support from Gobierno de Aragón through a doctoral fellowship.

## Appendix A: Complexity associated to the VQE

To compute $F$ we must express our *ansatz* as a unitary of the form $U = \mathcal{T}e^{-i\int_0^T \mathcal{H}(\tau)\,d\tau}$ where $\mathcal{H}$ is written in terms of Pauli matrices $\{\sigma_x, \sigma_y, \sigma_z\}$ and tensor products of these matrices. To do so, recall that the local VQE *ansatz* only contains one and two qubit gates (between nearest neighbors). To construct the effective Hamiltonian, notice that

$$R_y(\theta_i) = e^{-i\frac{\theta_i}{2}\sigma_y} \ . \tag{A1}$$

Now, the C-Z gate, can be decomposed

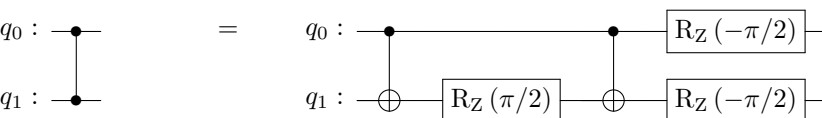

Therefore

$$C\text{-}Z = e^{-i\frac{\pi}{4}(\sigma_z^0\sigma_z^1 - \sigma_z^0 - \sigma_z^1)} = e^{-i\frac{\pi}{4}\sigma_z^0\sigma_z^1}e^{i\frac{\pi}{4}\sigma_z^0}e^{i\frac{\pi}{4}\sigma_z^1} . \tag{A2}$$

If we substitute in the representation of a layer of the ansatz, we find that each one of the building blocks marked with a dashed line in the main text is represented by a unitary of the form

$$U = e^{-i\sum_j \frac{\theta_j}{2}\sigma_y^j}e^{-i\frac{\pi}{4}(\sigma_z^0\sigma_z^1 - \sum_j \sigma_z^j)} \approx e^{-i(\sum_j \frac{\theta_j}{2}\sigma_y^j + \frac{\pi}{4}\sigma_z^0\sigma_z^1 - \frac{\pi}{4}\sum_j \sigma_z^j)} , \tag{A3}$$

Finally,

$$\mathcal{H} = \sum_j \frac{\theta_j}{2}\sigma_y^j + \frac{\pi}{4}\sigma_z^0\sigma_z^1 - \frac{\pi}{4}\sum_j \sigma_z^j . \tag{A4}$$

More generally, each layer of the ansatz can be written as an operator of the type

$$\mathcal{H} = \mathcal{H}_{\text{even}} + \mathcal{H}_{\text{odd}} , \tag{A5}$$

where

$$\mathcal{H}_{\text{even}} = \frac{1}{t}\left(\sum_i \frac{\theta_i}{2}\sigma_y^i - \frac{\pi}{4}\sum_{i=0}^{L-1}\sigma_z^i + \frac{\pi}{4}\sum_{i=\text{even}}\sigma_z^i\sigma_z^{i+1}\right) , \tag{A6}$$

$$\mathcal{H}_{\text{odd}} = \frac{1}{t}\left(\sum_{i=0}^{L-2} \frac{\theta_{i+L}}{2}\sigma_y^i - \frac{\pi}{4}\sum_{i=1}^{L}\sigma_z^i + \frac{\pi}{4}\sum_{i=\text{odd}}\sigma_z^i\sigma_z^{i+1}\right) . \tag{A7}$$

Now we use a Trotter decomposition to compute the complexity of this circuit. We have fixed the total evolution time to 1 and each layer is considered a Trotter step. This way, $t = T/\#\text{steps} = 1/d$, where d is the number of layers of the circuit. Now, using Eq. (6) we find

$$F(U) = \sqrt{\sum_i^{2(L-1)}\left(d\frac{\theta_i}{2}\right)^2 + 3(L-1)\left(d\frac{\pi}{4}\right)^2} . \tag{A8}$$

Here, $L-1$ corresponds to the number of C-Zs in the layer, with $L$ is the number of qubits. Now, the complexity is nothing but the integral of this functional across the number of layers in the circuit

$$\mathcal{C}_{\text{N}} = \int_0^1 F(U)dt \approx \sum_{j=1}^d F(U)\frac{1}{d} , \tag{A9}$$

which leads to Eq. (62) in the main text.

## Appendix B: Other paths in the adiabatic algorithm

In Sec. IV A we show an adiabatic evolution for the Transverse Field Ising model where we let the field fixed as we increase the interaction between the neighbouring spins. However, we could have let the interaction fixed and switched on the transverse field, going from a classical Ising model to the TFI. In Fig. 9 we show this possible adiabatic path. The behaviour of the gap between the ground state and the first excited state is qualitatively different, to the point of even closing. This results in a much worse performance for small values of the field.

Similarly, the gap behaviour also causes a big impact in the ZZXZ model. In Fig. 10 we show that for odd number of spins in the chain we get a higher complexity as the gap presents a dip at intermediate times which makes necessary longer times to achieve the fidelity threshold.

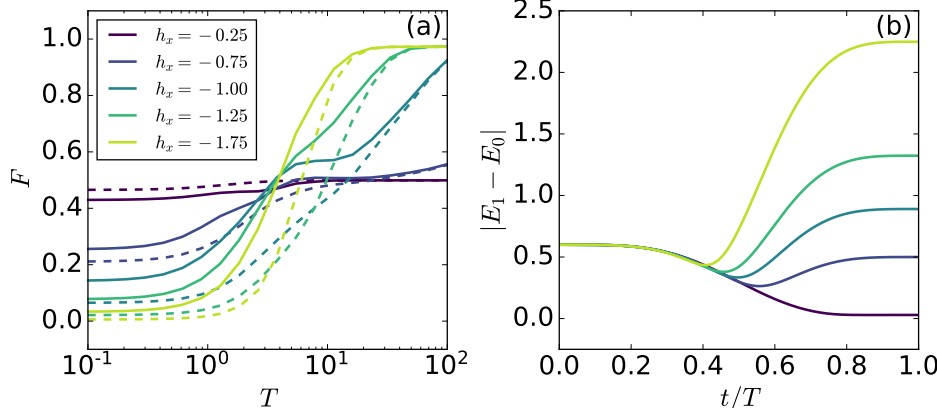

FIG. 9. Adiabatic evolution for the TFI model by switching on the transverse field instead of the spin-spin interaction. (a) Evolution of the fidelity obtained with shortcuts to adiabaticity (solid lines) and without them (dashed lines) for increasing time lengths of the full algorithm and $L = 12$. We see a clear difference with the plot in the main text, where the field is fixed and we vary the interaction, $J$. The gap closes much earlier for small field values (b), making the algorithm need much longer times to achieve high fidelity.

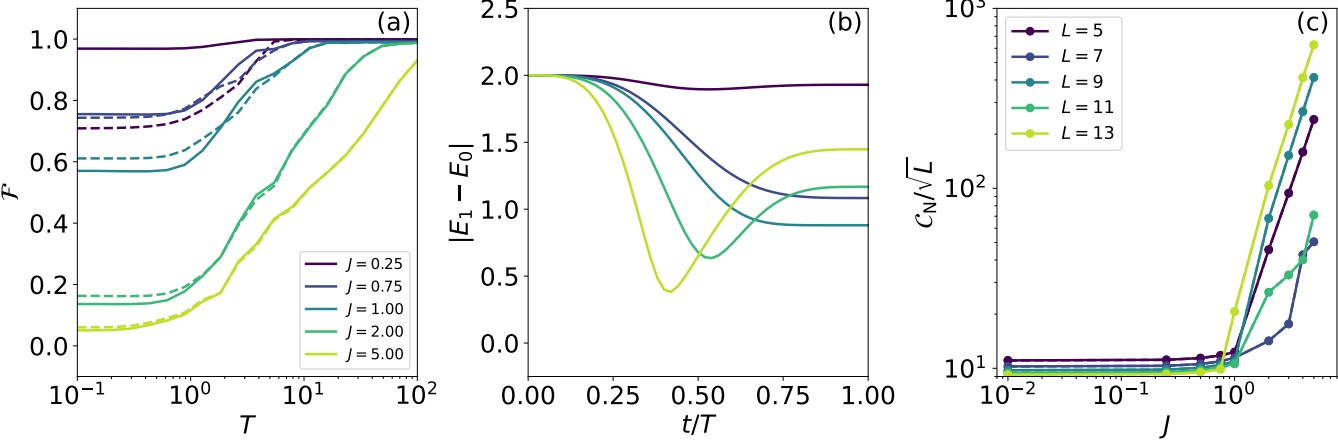

FIG. 10. Evolution in the ZZXZ model of the fidelity (a), the gap between the ground state and the first excited state (b) and the complexity (c) for spin chains with odd number of constituents. The dip at intermediate times in the gap causes the complexity to increase compared to the even case.

## Appendix C: Adiabatic dependence on the energy gap

In this appendix we show the dependencies obtained for the Nielsen complexity in the adiabatic algorithm for the two models studied: TFI and ZZXZ. The discussion in section IV A based on the adiabatic theorem tells us that $\mathcal{C}_N \sim T$ and we can expect the dependence between the preparation time, $T$, and the energy gap, $\Delta \equiv E_1 - E_0$, to be $T \sim \Delta^{-2}$ in the best case. Therefore, we now check whether, as the gap closes for increasing chain size, we observe a dependence $\mathcal{C}_N \sim \Delta^{-2}$.

In figures 11 (TFI model) and 12 (ZZXZ model) we can see that indeed, when $|J|$ is sufficiently small (before and just at the transition) the complexity seems to respond to this trend. However, when we enter the ordered phase (and the gap closes) we completely lose the predicted dependence. One of the main reasons may be the one described in the main text. That is, the preparation time is not determined with the necessary precision. On the other hand, it may simply be that the dependence is different.

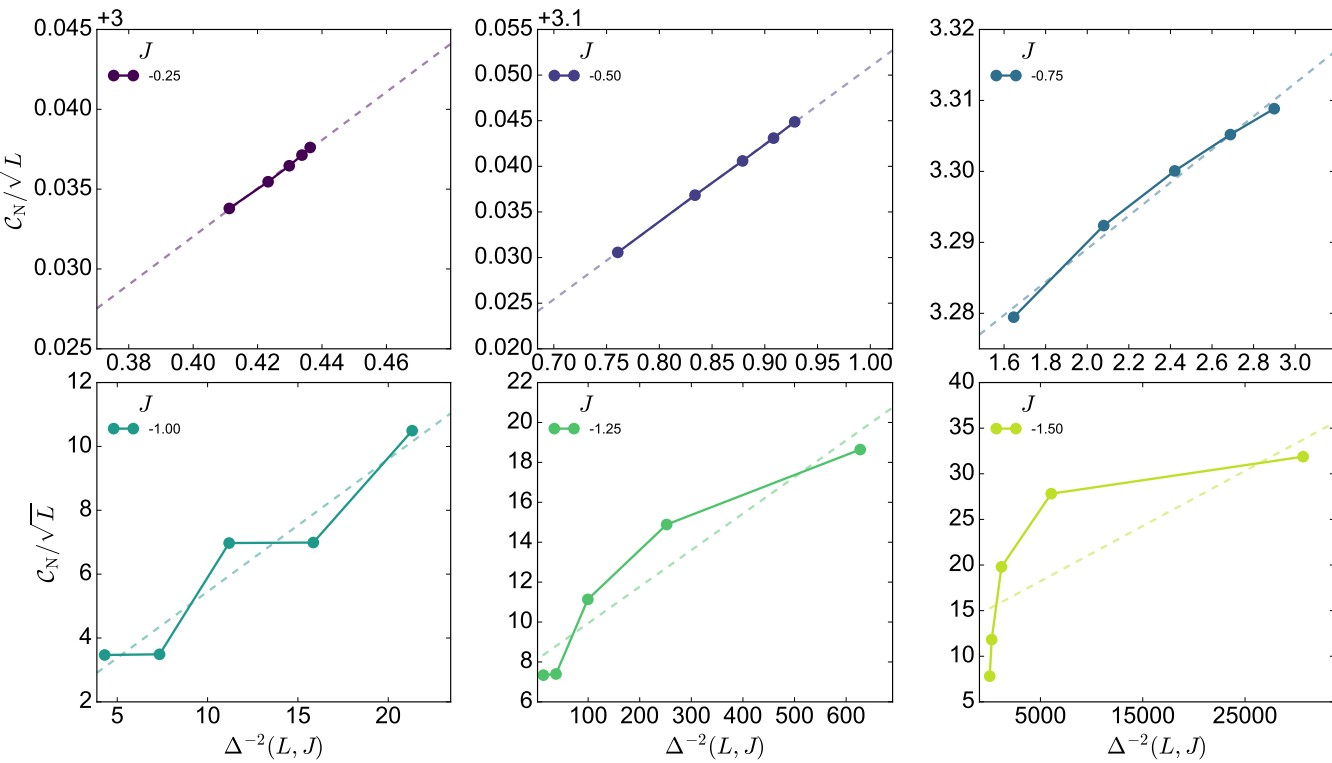

FIG. 11. Relation between the Nielsen complexity in the adiabatic preparation and the system gap for the TFI model.

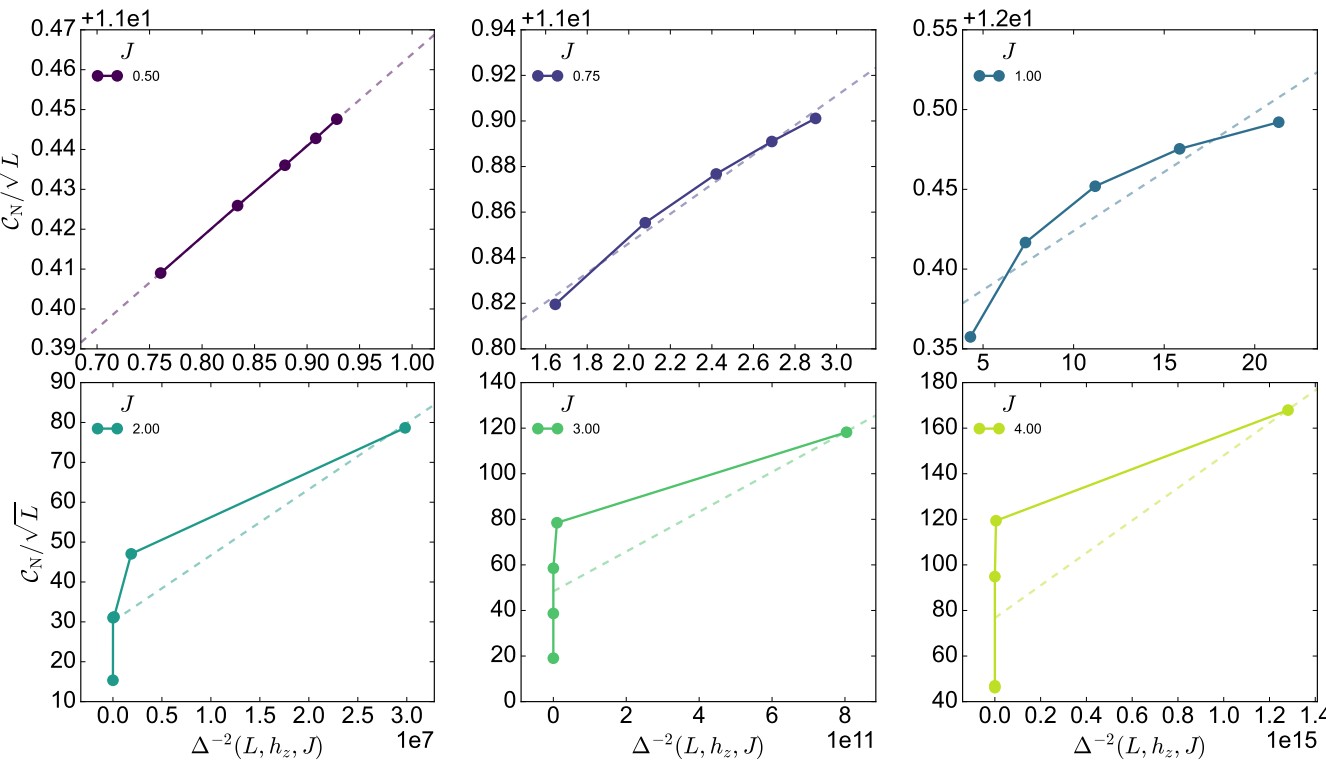

FIG. 12. Relation between the Nielsen complexity in the adiabatic preparation and the system gap for the ZZXZ model.

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
