# Peer review of "Circuit Complexity through phase transitions: consequences in quantum state preparation"

_SciPost Physics_

## Round 1 · Referee Report · Zakaria Mzaouali (Referee 1) · 2023-5-28

Strengths

The authors present an analysis of computational complexity in quantum algorithms and its behavior upon crossing a quantum phase transition in well-known quantum many-body systems. The paper is well written, the figures are clear, and the authors give a clear discussion and interpretation of their results.

Weaknesses

1) The introduction of the paper could be improved by providing further details on the notion of computational complexity and the challenges it imposes on emergent quantum technologies, i.e. quantum computing.

Report

An interesting result from the authors is the impact of the energy gap on computational complexity. It would be interesting to extend the present study to the case where the Hamiltonian is driven in a way that takes into account the spectral structure of the system. (see arXiv:2208.09271).

I recommend the publication of the manuscript after taking into account the requested changes

Requested changes

1) In the "Complexity through QPTs" subsection on page 7, there is a typo in the second line. I think the authors are referring to Figure 1 instead of Figure 6.

2) The authors should stick to one notation for the chain size, either $N$ or $L$, throughout the whole paper

  • validity: good
  • significance: good
  • originality: good
  • clarity: good
  • formatting: acceptable
  • grammar: good

Author:  Sebastián Roca-Jerat  on 2023-07-28  [id 3846]

(in reply to Report 1 by Zakaria Mzaouali on 2023-05-28)

Dear Referee,

We have carefully considered the comments and suggestions and modified the text accordingly.
More specifically, we have corrected the typo on page 7 as well as fixed the notation for the chain size.
Furthermore, we have added the reference included in the Report and integrated it into the discussion of the text at the beginning of page 15, where the effect of shorcuts to adiabaticity is addressed.

---

## Round 1 · Referee Report · Anonymous (Referee 2) · 2023-6-29

Strengths

1- the subject matter is quite interesting and timely 2- the first section contains a review of various concepts of complexity and their relations 3- the qualitative idea, a grower in complexity as the algorithm takes the system through a QPT, is certainly interesting

Weaknesses

1- the presentation is somewhat imprecise and too qualitative 2- the quantitative relation between complexity and gap closing though presented for FS complexity, where it is related to the scaling of the fidelity susceptibility studied in other contexts, it is hardly explored for Nielsen complexity

Report

The manuscript presents a study of circuit (and geometric) complexity of various quantum algorithms and unitary evolutions where the initial and target states are potentially separated by a quantum phase transition. The purpose of the manuscript is to show that the crossing of the quantum critical point leads to complexity increasing with system size. While such divergence is studied systematically for the Fubini-Study complexity where the scaling (dominated but the neighborhood of the QCP) is essentially connected to that of the fidelity susceptibility, for the Nielsen complexity the computation is done numerically for adiabatic algorithms and for VQE. The qualitative statement obtained is that whenever the algorithm is “aware” of a QCP the associated complexity is seen to increase and diverge with system size.

I think the subject matter of this paper quite interesting though at present the study seems more a collection of semi-qualitative observations rather than a systematic study of the relation between complexity and QCP. The introduction, section 1.A, is very nice though it would help to expand the explanation a bit, in particular between Eq.(11) and (14) where it is explained a connection between the two concepts. In particular I think it would be better not to confuse the reader and maybe use more space but state clearly the problem at all stages: the Nielsen complexity is related to transformations between two fixed initial and target state, the FS is a distance between states belonging to a given parametric manifold, the relation is given by such and such.

In the part concerning FS Complexity is not clear to me what new quantitative elements are brought in in this paper as compared to what was already know about fidelity susceptibilities though QCP (see e.g. the works of De Grandi, Gritsev and Polkovnikov a few years back). It would be useful to state it clearly.

In turn, the part concerning Nielsen complexity, which to my knowledge is novel, is extremely qualitative. Certainly it gives a hint towards interesting phenomenology but It would have been useful to have a study (even numerical) of scaling with systems size, how this is affected (or not) by the properties of the QCP, etc.

Overall I think that the paper with significant improvements in the presentation, in discussing the relation with previous literature and more quantitative statements in the final part could be published in SciPost Physics.

Requested changes

1- revise the section on various concepts of complexity improving clarity 2- discuss relation to fidelity susceptibility literature 3- make more quantitative statements in Nielsen complexity

  • validity: good
  • significance: good
  • originality: ok
  • clarity: low
  • formatting: excellent
  • grammar: good

Author:  Sebastián Roca-Jerat  on 2023-07-28  [id 3847]

(in reply to Report 2 on 2023-06-29)

Dear Referee,

Please see the attached file for the response.

Attachment:

complexity_author_response_2.pdf

---

## Editorial Decision

resubmitted